# Production planning of a furniture manufacturing company with random demand and production capacity using stochastic programming

José Emmanuel Gómez-Rocha[1], Eva Selene Hernández-Gress[2]* , Héctor Rivera-Gómez[1]

1 Engineering Academic Area, Universidad Autónoma del Estado de Hidalgo, Pachuca de Soto, Hidalgo, México, 2 School of Engineering and Science, Tecnologico de Monterrey, Pachuca de Soto, Hidalgo, México

☯ These authors contributed equally to this work.
* evahgress@tec.mx, evaselenehg@hotmail.com

## Abstract

In this article two multi-stage stochastic linear programming models are developed, one applying the stochastic programming solver integrated by Lingo 17.0 optimization software that utilizes an approximation using an identical conditional sampling and Latin-hyper-square techniques to reduce the sample variance, associating the probability distributions to normal distributions with defined mean and standard deviation; and a second proposed model with a discrete distribution with 3 values and their respective probabilities of occurrence. In both cases, a scenario tree is generated. The models developed are applied to an aggregate production plan (APP) for a furniture manufacturing company located in the state of Hidalgo, Mexico, which has important clients throughout the country. Production capacity and demand are defined as random variables of the model. The main purpose of this research is to determine a feasible solution to the aggregate production plan in a reasonable computational time. The developed models were compared and analyzed. Moreover, this work was complemented with a sensitivity analysis; varying the percentage of service level, also, varying the stochastic parameters (mean and standard deviation) to test how these variations impact in the solution and decision variables.

## 1. Introduction

Carrying out an aggregate plan is important in manufacturing industries, especially those where it is planned in periods of 3 to 18 months, or medium term. The production plan seeks to determine the optimal levels of production, hiring, layoffs, inventories, subcontracting, etc. This work presents an aggregate plan that was made for a company that manufactures furniture in the State of Hidalgo. Initially, a first approach to the solution of the problem was made in [1]. In this work, only the production capacity was considered as a random variable using two models, one with a continuous probability distribution and the other with a discrete one.

**Data Availability Statement:** The data and all of the statistical analysis underlying this study are available on Figshare (https://doi.org/10.6084/m9.

figshare.14444609.v1) along with the lingo models
(https://doi.org/10.6084/m9.figshare.14450430).

**Funding:** The author(s) received no specific
funding for this work.

**Competing interests:** The authors have declared
that no competing interests exist.

However, another extremely important random variable had been ignored due to complexity: demand. Therefore, the motivation for this work is to improve productivity, have efficient policies to manage its production and minimize production costs, developing models of aggregate production plans (APP) with uncertainty due to a real need of a furniture company, Models are considering real characteristics such as human factor, multi-period production criteria and service level policy due to the use of backlogs.

The main objective of this article is to develop a multi-state stochastic optimization model applied to an APP of a local company, where the production periods are defined as the states, the randomness of production capacity and demand are modeled through a continuous probability distribution using the stochastic programming solver integrated by Lingo. Two models are proposed, Model-I only could solve the problem for a maximum of three periods, due the complexity of using a continuous probability distribution, a second model is proposed with a discretization of the probability distributions (Model-II) which could solve the problem up to four periods. In both models a scenario tree is created. In general, this work compares the efficiency between Model-I and Model-II in resolution time, number of iterations, expected value (EV), wait-and-see value (WS), and expected value of perfect information (EVPI). The obtained results help to determine the advantages about the proposed model (Model-II) with respect to Model I and is useful to understand the scope of both models and in which cases it is advisable to use each one. In addition, both models consider the impact of the service level restriction on the optimal solution and what happen when parameters of the distribution probabilities are varying.

The novelty of this work could be summarized in five points,1) this study provides a mathematical programming model that has been adapted for real needs of a company, which incorporates a service level constraint that it is not found in the literature, usually a confidence percentage is used (which could turn the problem into a chance constraint programming. 2) In the literature only the expected value of the objective function is reported (here and now solution) with the history of the process considered, that is, with the nonanticipativity constraints or the value of the expected objective function. If these constraints are removed, the wait and see solution appears, in our research, both solutions are reported, also the absolute difference of the two solutions is reported, called the expected value of perfect information that could help the company to deal with uncertainty in economic decisions. 3) An extensive sensitivity analysis is carried out, varying the cost parameters, the percentage of the service level and varying the parameters of the probability distribution of uncertainty. Few studies carry out a sensitivity analysis, but in our knowledge, nobody analyzes the impact of service level and varies the parameters of the probability distribution. Through this sensitivity analysis, interesting results were obtained, for example, that the total cost of the APP goes down, when the variability of the production capacity (standard deviation of the probability distribution) is reduced. 4) Due to the complexity of the problem, the software could not solve the problem satisfactorily for a fourth period, finding a solution that is only feasible, then, a second model is developed using discretization of the probability distribution, it has been shown that if the distances of both distributions are minimal, the solution found is closer than the true optimum [2–4] the quality of the proposed model is presented in the results, where both models are compared.

Finally, 5) a methodology to deal with problems using stochastic programming is proposed, although it was applied to the case of this APP, can be implemented in other areas of industrial engineering sciences, such as supply chain networks, problems of vehicle routing, design, and redesign of layouts, among others. Advantages and disadvantages are detailed in the conclusions section. Table 1 shows a comparison between some relevant studies in the area and our study, so that the novelty and contribution of our proposal can be observed.

**Table 1. Studies about APP with uncertainty.** Own elaboration.

| Characteristics | This study | Kazemi et al. [24] | Jamalnia et al. [16] | Zhao et al. [18] | Tirkolaee et al. [21] |
|---|---|---|---|---|---|
| **Model Class** | Mixed Integer linear multi-stage stochastic | Mixed Integer linear multi-stage stochastic | Multi-objective nonlinear multi-stage stochastic | Mixed integer multi-objective nonlinear multi-stage stochastic | Mixed-integer multi-objective nonlinear multi-stage stochastic |
| **Source of uncertainty** | Demand and capacity production | Demand and yield | Demand | Patient recruitment | Demand and costs |
| **Probabilistic distribution for the source of uncertain** | Normal distribution and discrete approximation | Discrete approximation | Discrete empirical distribution | Poisson distribution | Fuzzy triangular number |
| **Number of models developed** | Two stochastic models | One stochastic model | One stochastic model | One stochastic model | One stochastic model |
| **Solution strategy** | Scenario tree approach using Lingo 17.0 | Scenario tree approach and scenario decomposition based in PHA using CPLEX 11 | Scenario decomposition based in relaxed nonanticipativity constraint using WWW-NIMBUS | Scenario tree approach and PHA using CPLEX 12.6 | Weighted goal programming using GAMS with CPLEX |
| **Level service** | Considered | Not considered | Considered | Not considered | Not considered |
| **Impact of level service** | Considered | Not considered | Not Considered | Not considered | Not considered |
| **Iterations required for a solution reported** | Considered | Not considered | Not considered | Not considered | Not considered |
| **CPU time required for a solution reported** | Considered | Not considered | Not considered | Considered | Considered |
| **Here and now solution** | Considered | Considered | Not considered | Considered | Considered |
| **Wait and see solution** | Considered | Not considered | Considered | Not considered | Not considered |
| **Expected value for perfect information** | Considered | Not considered | Not considered | Not considered | Not considered |
| **Sensitivity analysis varying *deterministic parameters*** | Considered | Not considered | Considered | Considered | Considered |
| **Sensitivity analysis varying *stochastic parameters*** | Considered | Not considered | Not considered | Not considered | Not considered |

The contribution of this work is a real problem where uncertainty affects the production system, generally, the models used in the literature consider demand as a random variable with a discrete approximation (one model), in this work, in addition, the human factor is considered as a stochastic parameter that can be modeled and 2 models are compared. The rest of the article consists of the following sections. Section 2 presents the literature review, section 3 describes the model under study, first as deterministic and then as stochastic. The methodology is introduced in section 4. The results are given in section 5. An extensive sensitivity analysis is performed in section 6. Section 7 concludes the article.

## 2. Literature review

Considering uncertainty within optimization problems remains a trending topic to be investigated because organizations face to stochastic variables when making decisions. A search was carried out in Scopus and in Web of Science written during 2020 and 2021 that used stochastic programming to solve optimization problems. Huang et. al [5] develop a multistage stochastic optimization model for system operators to efficiently schedule power-generation assets to co-optimize power generation and regulation reserve service under uncertainty. Ghayour et al. [6] present an approach called MLPR with linear programming used as its core in order to solve the influence maximization problem in the linear threshold model, that is one of two classic stochastic propagation models that describe the spread of influence in a network.

Robust Multi-product Newsvendor Model with Substitution, where the demand and the substitution rates are stochastic and are subject to cardinality-constrained uncertainty sets that is an NP hard problem is presented in [7].

Also, Basciftci et. al [8] reformulate the robust facility location problem, in which they interpret the moments of stochastic demand as functions of facility-location decisions. In Shone et. al [9], stochastic modeling applications within aviation are presented, with a particular focus on problems involving demand and capacity management and the mitigation of air traffic congestion; using operations research perspective, including analytical queueing theory, stochastic optimal control, robust optimization and stochastic integer programming. Ghasemi et. al [10] present an Evolutionary Learning Based Simulation Optimization (ELBSO) method embedded within Ordinal Optimization. In ELBSO a Machine Learning (ML) based simulation metamodel is created using Genetic Programming (GP) to replace simulation experiments aimed at reducing computation; ELBSO is evaluated on a Stochastic Job Shop Scheduling Problem (SJSSP). Zhang et. al [11], consider a stochastic vehicle routing problem with probability constraints; the probability that customers are served before their (uncertain) deadlines must be higher than a pre-specified target. Wang et. al [12] propose a model to solve a project scheduling problem where resource assignments and activity schedules need to be determined to achieve a set of due-date requirements as well as possible. Torres et. al [13] present multistage stochastic program for the design and management of flexible infrastructure networks with stochastic demands.

In the methods for solving stochastic programming, Dowson and Kapelevich [14] develop the Julia package for multistage stochastic and dual programming and Gangammanavar et. al [15] work with stochastic decomposition for two-stage stochastic linear programs with random cost coefficients.

Talking specifically of Aggregate planning, there are some works related with it, Jamalnia et al. [16] mention that the methodologies applied to deal with aggregate production plans (APP) under uncertainty can be classified into six main categories: stochastic mathematical programming, possibilistic programming, fuzzy mathematical programming, simulation modelling, metaheuristics, and evidential reasoning. Here are some important works.

Using Multi objective stochastic optimization, Nowak [17] introduced a work that combines linear mathematical programming with multiple objectives (multi-objective), simulation and an interactive approach with uncertain demand. In contrast, Jamalnia et al. [16] presented a nonlinear stochastic optimization model with multiple objectives for an aggregate production plan under uncertainty. The WWW-NIMBUS software was used, using more than 500 decision variables and 1,000 restrictions with 7 objectives. Zhao et al. [18], showed a case study for the pharmaceutical industry where they optimize the quantity of production, minimizing the duration of clinical trials and operating costs. The problem is formulated by the multistage stochastic programming using C# programming language as well as CPLEX as a solver, comparing the Progressive Hedging Algorithm (PHA) and the Sample Average Approximation (SAA) to test the optimality GAP, the time to solve a scenario and period, the memory used, among others. Rakes, Franz, and Wynne [19] and Chen and Liao [20] also submitted works where there is uncertainty in production plans with multiple objectives. Recently, Tirkolaee et al. [21] investigates a novel fuzzy multi-objective multi-period Aggregate Production Planning (APP) problem under seasonal demand. As two of the main real-world assumptions, the options of workforce overtime and outsourcing are studied in the proposed Mixed-Integer Linear Programming (MILP) model.

Stochastic optimization has been applied in various production plans where the environment tends to have uncertainty as mentioned by Birge and Louveaux [22]. Wagner and Whitin [23] are considered the first to study production plans under uncertainty, solving a one-state

problem through dynamic forward programming for a single product. Kazemi et al. [24] perform a multi-stage stochastic mixed integer linear programming model for sawmill production (lumber industry) in Canada; giving a solution through an approximation by decomposition of scenarios. CPLEX software was used for the solution. It was shown that problems with more than 22,000 restrictions and 44,000 decision variables can be solved within a reasonable time, this was a single objective problem. Huang [25] also presented models for production plans through multi-stage stochastic mathematical programming.

Using, nonlinear stochastic optimizatiom, Ning et al. [26], Mirzapour Al-e-hashem et al. [27] and Lieckens and Vandaele [28] developed mixed integer nonlinear mathematical programming methodologies to study the decision problem of an aggregate production plan and they consider demand and lead time as variables with uncertainty in their proposals. Nasiri et al. [29] suggested a non-linear stochastic model for a production and distribution plan in a three-level supply chain (suppliers, production centers and customers). This model was solved by using commercial packages such as Lingo in addition to a heuristic programmed in Matlab. Similarly, Nasiri et al. [30] proposed a nonlinear stochastic integer mixed mathematical programming model for a supply chain to later extend their work in a multi-stage system [31]. Ning et al. [26] presented a multi-product nonlinear application model where market demand and production cost are uncertain.

Robust optimization and different techniques under uncertainty have been used in research such as Leung and Wu [32], Kanyalkar and Adil [33], Mirzapour Al-e-hashem, Malekly and Aryanezhad [34], Mirzapour Al-e-hashem, Aryanezhad and Sadjadi [35], Makui et. al [36] used robust optimization techniques to deal with aggregate production plans under uncertainty.

A key concept in stochastic programming is *scenario tree*. Kazemi et al. [24] mentioned that a scenario tree is a computationally viable way of discretizing the dynamic stochastic data underlying a problem over time, this allows to reach viable solutions in reasonable times. In a competitive environment it is required to have results in faster times. Hu and Hu [37] used a scenario tree to optimize the job shop scheduling problem (JSSP) and the economic order quantity (EOQ) under uncertain demand, Körpeoglu, Yaman and Aktürk [38] used it to solve the master production scheduling problem (MPS). In Table 1 the studies that closer to our proposal are compared in different characteristics.

Reviewing the works found that are related to stochastic programming, it was observed that all of them use mixed integer programming, Jamalnia et al. [16], Zhao et al. [18] and Tirkolaee et al. [22] with nonlinear multiobjective, while Kazemi et al. [24] uses multistage stochastic programming which is the same as that used in our proposal. Tirkolaee et al. [22] use demand and costs as stochastic variables, while Kazemi et al. [24] demand and yield; the others, only use a single variable as a stochastic. In their approaches, a single approximation is used to explain the uncertainty, which is discrete, that is, a single stochastic model, and in our case two models, that allow us to compare between the normal distribution and its discretization in order to offer to the company a good solution in a reasonable computational time. The proposals found use the scenerio tree, except Tirkolaee et al. [22] that deal with the problem with weighted goal using GAMS. The level of service is only handled in Zhao et al. [18], but the impact of the service level on the solution is not considered. The sensitivity analysis varying stochastic parameters was not used in the approaches found. After analyzing the characteristics of the studies found, the gap was in considering the level of service, which was a very important restriction for the company in the case study, considering demand and labor as stochastic variables, which were two variables that generate a lot of uncertainty in the company and also varying the stochastic parameters. Finally, generate an efficient model, that is to say, a model that obtains a good response in a reasonable computational time, for that reason 2 models were

tested for comparison. In the next section the development of mathematical models will be described.

## 3. Development of mathematical models

To develop the stochastic programming models a proposed methodology was used, where before developing the stochastic programming models directly, a first deterministic model was created, in order to observe the behavior of the model and after extended it by incorporating the stochastic parameters.

### 3.1 Deterministic model

The deterministic base model for the aggregate production plan for optimizing the total cost of production, determines the number of workers that must be hired and fired per month, the number of parts that will be produced, the parts that will be sent to the finished product warehouse (inventory) and the backlog per month with a service level of 90% each period. The model considers the following assumptions that are considered in the literature review and some others are specific for the company:

- The demand is known for all periods.

- The production capacity per worker is the same for all months.

- Backlog may exist in the company, but it is penalized for missing units, this is because there is no possibility of buying parts (outsourcing) and overtime is not allowed in the company.

- Backlogged parts must be satisfied in the next period.

- The costs associated with the production of a part, inventory and backlogs are linear.

- There is enough raw material for production of parts ($X_t$).

The notation used for the model is shown in the following Table 2. Where the indices, parameters and decision variables are detailed, later the objective function and the model restrictions are explained.

The deterministic base model used was based by the one in Ravindran and Warsing [39] adapting it to the needs of the company by incorporating the service level and minimum inventory restrictions. The objective function of the deterministic model in Eq (1), minimizes the total cost of the APP, considering the costs associated with the workers assigned to production, fired, hiring, as well as inventory costs, backlogs and production, as defined in the following equation:

$$min\ Z = \sum_{t=1}^{T} P_t C_P + \sum_{t=1}^{T} F_t C_F + \sum_{t=1}^{T} R_t C_R + \sum_{t=1}^{T} I_t C_I + \sum_{t=1}^{T} X_t C_X + \sum_{t=1}^{T} S_t C_S \quad (1)$$

The model constraints are the following:

$$W_t = W_{t-1} + R_{t-1} - F_{t-1} \forall t = 2, \ldots, T \quad (2)$$

$$W_t = P_t + F_t \forall t = 1, \ldots, T \quad (3)$$

$$X_t + I_{t-1} = D_t + S_{t-1} + I_t - S_t \forall t = 1, \ldots, T \quad (4)$$

$$X_t \leq kP_t \forall t = 1, \ldots, T \quad (5)$$

$$I_t \geq I_\alpha \forall t = 1, \ldots, T \quad (6)$$

**Table 2. Notation of the deterministic model (3.1).**

| Indexes: | |
| --- | --- |
| $T$ | Time horizon of the aggregate plan $t \in T$ |
| **Parameters:** | |
| $D_t$ | Monthly product demand for period $t$ |
| $C_P$ | Salary of the production workers per month |
| $C_F$ | Cost of firing a worker |
| $C_R$ | Cost of hiring a worker |
| $C_I$ | Inventory cost per part |
| $C_S$ | Backlog cost per part |
| $C_X$ | Unit cost of producing a part (raw material and indirect cost) |
| $I_\alpha$ | Minimum inventory determined by company policies |
| $k$ | Monthly production capacity per worker |
| **Decision variables:** | |
| $W_t$ | Number of workers per month |
| $P_t$ | Number of workers assigned to production per month |
| $R_t$ | Number of hired workers per month |
| $F_t$ | Number of fired workers per month |
| $X_t$ | Number of parts to be produced per month |
| $I_t$ | Number of parts in the inventory per month (with $I_0$ known) |
| $S_t$ | Backlog per month (with $S_0 = 0$) |

$$D_t - S_t \geq .90 D_t \forall t = 1, \ldots, T \tag{7}$$

$$W_t, P_t, R_t, F_t, X_t, I_t, S_t, D_t \geq 0 \forall t = 1, \ldots, T, t \in T \tag{8}$$

$$P_t, F_t, X_t, I_t, D_t \in \mathbb{R}_+ \times \mathbb{Z}_+ \forall t = 1, \ldots, T \tag{9}$$

Constraint (2) focuses on the size of the workforce in the company, and it indicates that the total number of workers in period $t$, it must be equal to those existing in period $t-1$, plus those hired in period $t-1$, minus those laid off in period $t-1$. Constraint (3) is about the allocation of the workforce; it simply defines how many workers will be assigned to production and the number of workers that will be laid off in period $t$. Constraint (3) complements the balance over the number of workers denoted in Eq (2) and specifies the assignment of workforce in production and the number of workers to be fired each month. Constraint (4) refers to the balance of demand and inventory in the company, where what is produced in period $t$ plus the inventory of period $t-1$, it must be equal to the demand of period $t$, plus the backlog of period $t-1$ plus the inventory in period $t$ minus the backlog of period $t$. Constraint (5) addresses the production capacity; it ensures that the workers assigned to production can manufacture the units required in period $t$. Constraint (6) defines that the inventory of the period is greater than the safety inventory defined by the company. Constraint (7) indicates that the service level is greater than or equal to 90% per month, this restriction is a company's policy. The expression (8) defines the constraint of non-negativity decision variables. The expression (9) specifies that the decision variables must be integers, which implies a mixed integer linear stochastic programming model.

The deterministic model is the basis for developing Model-I and Model-II, both are of the multi-stage stochastic programming type, where production capacity and demand are random parameters. The Model-I associates these random parameters to a normal distribution, so the solution strategy is to use the stochastic programming solver integrated in Lingo 17.0 for Model-I, which internally creates a scenario tree with the same probability of occurrence for each scenario. Subsequently, a Model-II is proposed through a discretization of the probability distributions to create a scenario tree, each with different probabilities.

Model-I has the same assumptions as the base model, in addition to the following:

- Demand follows a normal distribution with mean 353 and standard deviation 29 and production capacity a normal distribution with mean 12 and standard deviation 2. These values were obtained through historical data of the company during 3 years prior to the start of the study using 40 data. The Arena ™ Input Analyzer was used to determ|wine these values. The statistical tests made to the data are shown complete in https://doi.org/10.6084/m9.figshare.14444609.v1.

- When considering a random demand, it is not possible to know how much to produce before the realization of this random event, but it must be determined how many workers are needed in each period. Therefore, the decisions of the same period can be made in different states. In Table 3 there are the parameters associated with this model:

Model-II has the same assumptions as the base model, in addition to the following:

- The discrete probability distribution is obtained with the maximum likelihood values calculating the probability of some defined intervals, then the Gaussian quadrature method is applied to calculate the probabilities (areas) of the intervals.

- The possible values of the random event demand are 324, 353 and 382 with probabilities 0.25, 0.50 and 0.25, respectively.

- The possible values of the random event production capacity are 10, 12 and 14 with probabilities 0.267, 0.466 y 0.267, respectively.

- Similar to the Model-I, some decision variables of the same period could be done in different state due the stochastic process.

In both cases, because the models use a scenario tree, the equivalent deterministic model for each problem consist of the construction of a large mixed integer linear programming

**Table 3. Parameters in Model I.** Own elaboration based in the organization information.

| Model I-Parameters: | |
| --- | --- |
| $D_t^\delta$ | $D_t^\delta \sim N(353, 29)$ |
| $C_P$ | $7000 |
| $C_F$ | $10000 |
| $C_R$ | $5000 |
| $C_I$ | $7 |
| $C_S$ | $65 |
| $C_X$ | $200 |
| $I_\alpha$ | 100 (Units). |
| $k_t^\omega$ | $k_t^\omega \sim N(12, 2)$ |

**Table 4. Notation of the stochastic model (11).**

| Indices: | |
|---|---|
| $T$ | Time horizon of the aggregate plan $t \in T$. |
| $H$ | Set of states (con $h \in H \ \forall \ h = 0, \ldots, H$). |
| $\Omega$ | Set of random events with $\{\omega, \delta\} \in \Omega$. |
| $\omega$ | Random event of production capacity. |
| $\delta$ | Random event of demand. |
| $N$ | Number of scenarios for capacity production with $N = i^t$. |
| $M$ | Number of scenarios for demand with $M = j^t$. |
| $i$ | Possible scenarios for capacity production with $i = \{10, 12, 14\}$ and $i \in \omega$. |
| $j$ | Possible scenarios for demand with $j = \{324, 353, 382\}$ and $j \in \delta$. |

| Parameters: | |
|---|---|
| $D_t^j$ | Monthly product demand for the scenario of each realization, with three possible values j = {324, 353, 382} and respectively probabilities $P^j$ = {0.25, 0.50, 0.25} (stochastic parameter) |
| $C_P$ | Salary of the production workers per month = $7000 |
| $C_F$ | Cost of firing a worker = $1000 |
| $C_R$ | Cost of hiring a worker = $5000 |
| $C_I$ | Inventory cost per part = $7 |
| $C_S$ | Backlog cost per part = $65 |
| $C_X$ | Unit cost of producing a part (raw material and indirect cost) = $200 |
| $I_\alpha$ | Minimum inventory determined by company policies = 100(units) |
| $k_t^i$ | Monthly production capacity per worker for the scenario of each realization, with three possible values $i$ = {10, 12, 14} and respectively probabilities $P^i$ = {0.267, 0.466, 0.267} (stochastic parameter). |
| $P^{ij}$ | Probability of occurrence of random events $\{\omega, \delta\}$, where the following equation is obtained: $$\sum_{j=1}^{M} \sum_{i=1}^{N} P^{ijt} = 1 \qquad (10)$$ Which indicate that the probability sum of all scenarios across the scenario tree must be one as indicated Eq (10). |

| Decision variables: | |
|---|---|
| $W_{t(h)}^{ij}$ | Number of workers per month |
| $P_{t(h)}^{ij}$ | Number of workers assigned to production per month |
| $R_{t(h)}^{ij}$ | Number of hired workers per month |
| $F_{t(h)}^{ij}$ | Number of fired workers per month |
| $X_{t(h)}^{ij}$ | Number of parts to be produced per month |
| $I_{t(h)}^{ij}$ | Number of parts in the inventory per month (with $I_0$ known). |
| $S_{t(h)}^{ij}$ | Backlog per month (with $S_0 = 0$) |

problem, where at the end, the probability of the sum of each scenario should be one as indicated Eq (10).

The notation for Model-II is detailed in Table 4, Model-I is similar in structure to Model-II, the difference is that Model-I, the probability of occurrence for each scenario is the same, in addition the occurrence values are obtained by Lingo using identical conditional sampling and for this, it uses variance reduction techniques to improve the quality of the solution.

The objective function of the stochastic Model-II (11), as in (1) minimizes the total cost of the APP, considering the costs associated with the workers assigned to production, fired

workers, hired workers, in addition to inventory costs, backlogs and production.

$$
\begin{aligned}
min\ z &= \sum_{j=1}^{M}\sum_{i=1}^{N}\mathrm{p}^{ijt}\sum_{t=1}^{T}P_{t(h-1)}^{ij}(\Omega)C_P + \sum_{j=1}^{M}\sum_{i=1}^{N}\mathrm{P}^{ijt}\sum_{t=1}^{T}F_{t(h-1)}^{ij}(\Omega)C_F + \sum_{j=1}^{M}\sum_{i=1}^{N}\mathrm{P}^{ijt}\sum_{t=1}^{T}R_{t(h-1)}^{ij}(\Omega)C_R + \sum_{j=1}^{M}\sum_{i=1}^{N}\mathrm{P}^{ijt}\sum_{t=1}^{T}I_{t(h)}^{ij}(\Omega)C_I \\
&+ \sum_{j=1}^{M}\sum_{i=1}^{N}\mathrm{P}^{ijt}\sum_{t=1}^{T}X_{t(h)}^{ij}(\Omega)C_X \\
&+ \sum_{j=1}^{M}\sum_{i=1}^{N}\mathrm{P}^{ijt}\sum_{t=1}^{T}S_{t(h)}^{ij}(\Omega)C_S
\end{aligned} \tag{11}
$$

The model constraints of the stochastic model are the following:

$$
W_{1(0)}^{ij} = P_{1(0)}^{ij} + F_{1(0)}^{ij} \tag{12}
$$

$$
X_{1(1)}^{ij} + I_0 = D_1 + I_{1(1)}^{ij} - S_{1(1)}^{ij} \tag{13}
$$

$$
X_{1(1)}^{ij} \leq k_1^{ij}P_{\mathbf{1}(0)}^{ij} \tag{14}
$$

$$
W_{t(h)}^{ij}(\Omega) = W_{t-1(h-1)}^{ij}(\Omega) + R_{t-1(h-1)}^{ij}(\Omega) - F_{t-1(h-1)}^{ij}(\Omega)
$$

$$
\forall t = 2,\ldots,T, h = 1,\ldots,H, \omega = 1,\ldots,N, \delta = 1,\ldots,\mathrm{M} \tag{15}
$$

$$
W_{t(h)}^{ij}(\Omega) = P_{t(h)}^{ij}(\Omega) + F_{t(h)}^{ij}(\Omega)
$$

$$
\forall t = 2,\ldots,T, h = 1,\ldots,H, \omega = 1,\ldots,N, \delta = 1,\ldots,\mathrm{M} \tag{16}
$$

$$
X_{t(h)}^{ij}(\Omega) + I_{t-1(h-1)}^{ij}(\Omega) = D_t^j + S_{t-1(h-1)}^{ij}(\Omega) + I_{t(h)}^{ij}(\Omega) - S_{t(h)}^{ij}(\Omega)
$$

$$
\forall t = 2,\ldots,T, h = 1,\ldots,H, \omega = 1,\ldots,N, \delta = 1,\ldots,\mathrm{M} \tag{17}
$$

$$
X_{t(h)}^{ij}(\Omega) \leq k_t^{ij}P_{t(h)}^{i}(\Omega)\forall t = 2,\ldots,T, h = 1,\ldots,H, \omega = 1,\ldots,N, \delta = 1,\ldots,\mathrm{M} \tag{18}
$$

$$
I_{t(h)}^{ij}(\omega) \geq I_{\alpha}\forall t = 1,\ldots,T, h = 1,\ldots,H, \omega = 1,\ldots,N, \delta = 1,\ldots,\mathrm{M} \tag{19}
$$

$$
D_t^j - S_{t(h)}^{ij}(\Omega) \geq .90D_t^j\ \forall t = 1,\ldots,T, h = 1,\ldots,H\ \omega = 1,\ldots,N, \omega \in \Omega, \delta = 1,\ldots,\mathrm{M} \tag{20}
$$

$$
W_1^{ij}, P_1^{ij}, R_1^{ij}, F_1^{ij}, X_1^{ij}, I_1^{ij}, S_1^{ij} \geq 0 \tag{21}
$$

$$
P_1^{ij}, R_1^{ij}, F_1^{ij}, S_1^{ij}, I_1^{ij} \in \mathbb{R}_+ \times \mathbb{Z}_+ \tag{22}
$$

$$
W_{t(h)}^{ij}(\Omega), P_{t(h)}^{ij}(\Omega), R_{t(h)}^{ij}(\Omega), \quad F_{t(h)}^{ij}(\Omega), \quad X_{t(h)}^{ij}(\Omega), \quad I_{t(h)}^{ij}(\Omega), \quad S_{t(h)}^{ij}(\Omega) \geq 0
$$

$$
\forall t = 1,\ldots,T, h = 1,\ldots,H, \omega = 1,\ldots,N, \delta = 1,\ldots,\mathrm{M} \tag{23}
$$

$$P_{t(h)}^{ij}(\Omega), R_{t(h)}^{ij}(\Omega), F_{t(h)}^{ij}(\Omega), S_{t(h)}^{ij}(\Omega), I_{t(h)}^{ij}(\Omega) \in \mathbb{R}_+ \times \mathbb{Z}_+ \forall t = 1, \ldots, T, h = 1, \ldots, H \quad (24)$$

$$\sigma_h = \mathscr{P}_h = \mathscr{q}_h = \sigma'_h = \mathscr{P}'_h = \mathscr{q}'_h = \sigma''_h = \mathscr{P}''_h = \mathscr{q}''_h \forall [h] = [0], \ldots, [H] \quad (25)$$

Constraints (12) and (14) are zero state (or first state) constraints, these are considered before the occurrence of the random event and are equivalent to the constraints (3) and (5) of the deterministic model. Constraint (13) is equivalent to constraint (4) only for the first period. Constraints (15) to (20) are the same of (2) to (7) of the deterministic model, considering the random events ω, δ the necessary corrections are made in order to deal with the uncertainty, these variables are called recourse variables.

Constraints (21) and (22) indicate the nonnegativity constraint and assigning some variables as integers for the zero state, it should also be noted that not all variables are forced to be integers, a computational advantage is obtained and makes the model of the mixed integer-linear class. Constraints (23) and (24) indicate the non-negativity constraint and assigning some variables as integers for the next states. The nonanticipativity constraints (25) with $\sigma_h, \mathscr{P}_h, \mathscr{q}_h, \sigma'_h, \mathscr{P}'_h, \mathscr{q}'_h, \sigma''_h, \mathscr{P}''_h$ y $\mathscr{q}''_h$ column vectors of the decision variables for every realization of ij (that is the reason to use 9 vectors) and $[h] = [0], \ldots, [H]$ the history process, where $\sigma$, $\mathscr{P}$ and $\mathscr{q}$ are the column vectors for the decision variables in the scenarios for the production capacity production low, medium and high respectively, and it is indicated if the demand is low when they do not have a quote, medium demand if they have one quote $(\sigma'_h, \mathscr{P}'_h, \mathscr{q}'_h)$ and double quote if the demand is high (, $\sigma''_h, \mathscr{P}''_h$ y $\mathscr{q}''_h$).

In Fig 1 it can be observed how for a period nine scenarios are generated considering 3 possible scenarios for each random variable, each one with its respective probability. Then for the first node another nine scenarios are generated, that is, for two periods the scenario tree grew up to a total of 81 scenarios. The sum of all the probabilities of the nine scenarios must be equal to the probability of the predecessor node to those scenarios, and in the same way the sum of all the 81 scenarios must be 1 and the filtering process of the sigma algebras is respected, which implies respecting the nonanticipativity constraints.

## 4. Methodology

The methodology used in this article was divided into five stages. Each stage is briefly described below:

1.- Data collection and analysis: In order to prepare the aggregate production plan, historical company data was used to obtain production parameters and associated costs. An ABC classification of products was carried out, where the main product was determined, this product generates more income for the company. In this case it turned out to be a rustic chair.

2.- Deterministic mathematical modeling: a deterministic model was developed that minimized the production costs of the product. The company's policies to maintain a minimum inventory every month and have a service level of at least 90%, both were considered in the model.

3.- Generation of the first stochastic model (Model-I): Due to the hiring and firing policy, ergonomic factors, learning curve, among other factors (elements, aspects), it was noted that the worker's production capacity was not constant over time. Based on historical production data, the Input Analyzer tool of the Arena ™ software was used, observing that the production capacity follows a normal distribution with a mean of 12 units and a standard

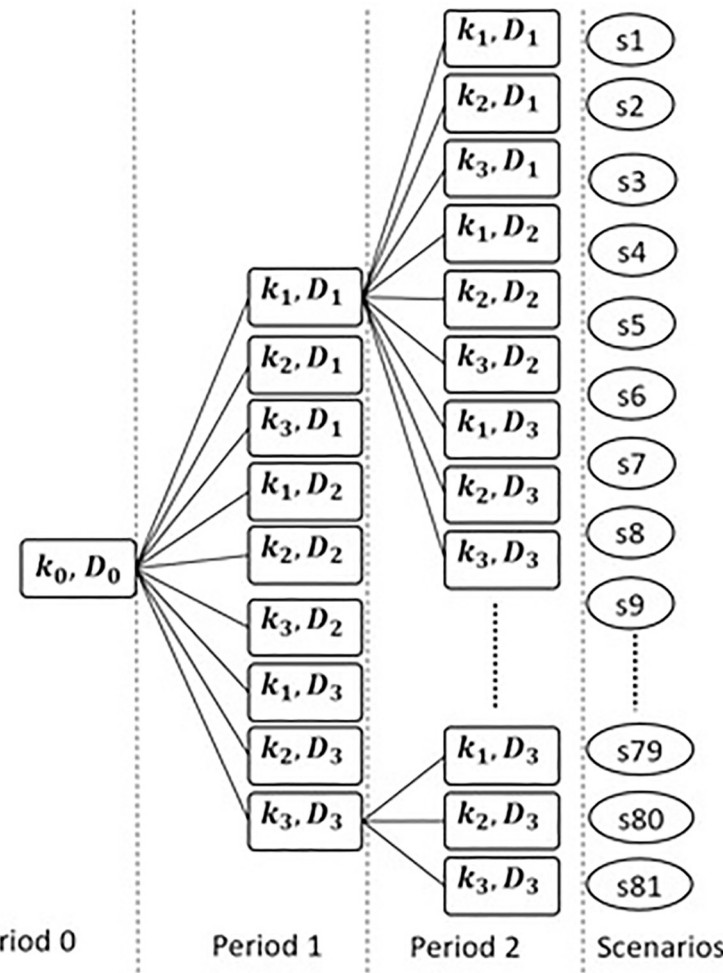

**Fig 1. Scenario tree for stochastic Model-I and Model-II.** Own elaboration.

deviation of approximately 2 units. Similarly, with a historical data on the demands of past periods, the Input Analyzer tool of the Arena ™ software was used, observing that the demand follows a normal distribution with a mean of 353 units and a standard deviation of approximately 29 units. Addtitionally, different test were made in order to validate the data, these analyzes are save in https://doi.org/10.6084/m9.figshare.14444609.v1.

The solution strategy for this problem was to implement the Monte Carlo technique of identical conditional sampling. Lingo uses identical conditional sampling and calculates the expectation for each state in a similar way to SAA, associating the random variable "Production capacity" and "demand" to a normal distribution with known distribution parameters. Additionally, the variance reduction technique is carried out using Latin-hyper-cubes in order to improve the solution.

The sample size is obtained in such a way that it does not generate an excessive computational effort in time and iterations and that in the solution obtained is still of quality. However, the use of variance reduction techniques in numerous studies (Kleywegt et al. [40]; Verweij et. al [41]; Shapiro & Homem-de-Mello [42]; Ruszczynski & Shapiro [43]) has shown that the number of samples to require a solution with the same quality as using a simple sampling is 10 to 100 times less, so the use of 3 samples is justified in this case.

4.- Generation of the second stochastic model (Model-II): The maximum likelihood values of the normal distribution were used to approximate it to a discrete distribution with three possible values 10, 12 and 14 (low, medium and high) with their probabilities obtained using the Gaussian quadrature technique [44] associated with that values (0.267, 0.466 and 0.267) for the random parameter "production capacity". For the random parameter "demand", the values 324, 353 and 382 with their respective probabilities (0.25, 0.50, 0.25) were used.

5.- Resolution and comparison of the models: Both stochastic models were processed to solve from two periods to the maximum number of periods that the computer was able to process. An optimization gap between the models was calculated. Also, a sensitivity analysis was performed, the impact of the percentage of service level and the parameters of probability distributions were analyzed.

Once the robustness and structure of the solution obtained has been confirmed, finally the optimal values of the related aggregate plan production will be prepared to be implemented in the manufacturing system. It is worth mentioning that this phase will be outside the domain of this study and remains as a possible future research direction. The lingo models are in https://doi.org/10.6084/m9.figshare.14450430. The work methodology is summarized in the Fig 2, in a flow chart.

## 5. Results

For the solution of the problem, a Dell Inspiron 5570 computer with an Intel® Core™ i5-8250U processor with 1.6GHz, with a 4Gb RAM memory with Windows 10 operating system was used. The Lingo 17.0 software was run using the Lingo stochastic solver, this solver uses an identical Monte Carlo sampling and Latin-hyper-cubes techniques to reduce the variance of the samples when a continuous distribution is used.

When the discrete distribution was implemented, only the empirical distribution and its probabilities were indicated to Lingo. It should be mentioned that in both cases the advantage that Lingo has is that by indicating the model as stochastic, it is unnecessary to indicate nonanticipativity restrictions, Lingo performs them internally.

For the solution of the equivalent determinists, five of the six available cores of the parallel processor were used (this option can be registered in the Lingo options), for the prior relaxation of the problem, it was specified in the linear solver to use 4 cores, each one with different strategies to solve the problem (two primaries using the simplex method, one barrier and the other dual).

For the mixed integer pre-solver, the maximum amount of heuristics (100) that Lingo has enabled were used to find optimal pre-solutions, the other capabilities were left as they come by default in Lingo. Finally, the Branch-and-Bound (B-and-B) algorithm was applied in the integer solver with Lingo's default criteria. In the case where a discrete distribution is used, a matrix decomposition was tested to perform a faster solution time.

Approximate results were found regarding the objective function of Model-I with respect to Model-II, which shows that the approximation of the continuous model with the discrete model demonstrated to be efficient when the problem becomes larger. The following comparative table shows how the problem grows when more periods are added in the aggregate production plan. This growth is because the model must generate the equivalent deterministic models for each branch of the scenario tree.

Table 5 shows how the problem grows as the number of periods increases, making it more difficult to find a solution to the problem if it is assumed that now the random variables are

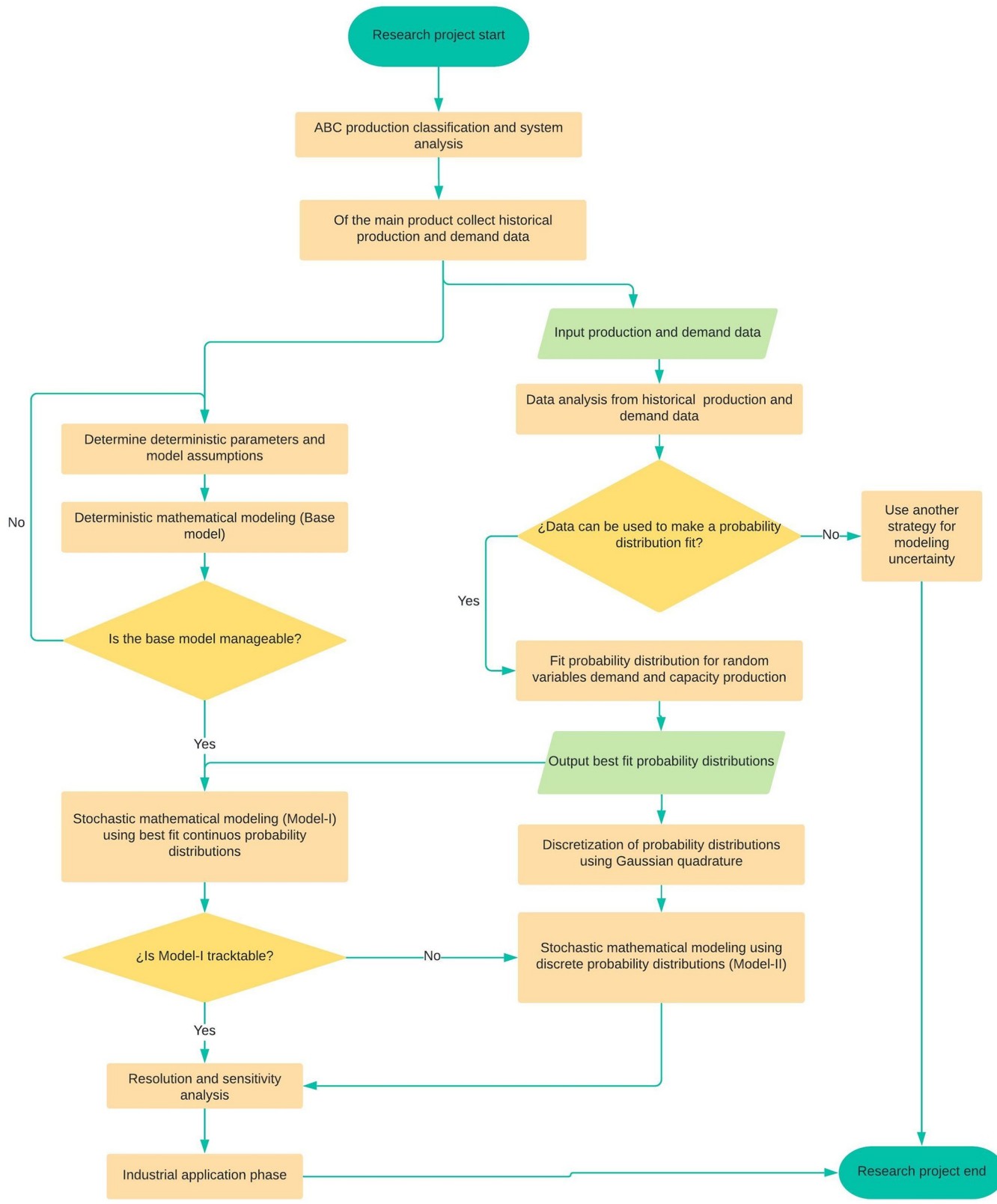

**Fig 2. Flow chart of the methodology.** Own elaboration.

**Table 5. Comparison of how the size of the problem increases with respect to the number of periods for Model-I and Model-II (R.V. denotes random variables, Var. denotes variables, and Int. Var. denotes integer variables).** Own elaboration.

| Periods | Scenarios | RV | Deterministic model | | | Equival Equivalent deterministic | | |
|---|---|---|---|---|---|---|---|---|
| | | | Var. | Int. Var. | Constraint | Var. | Int. Var. | Constraints |
| 2 | 81 | 2 | 14 | 10 | 12 | 1134 | 810 | 1876 |
| 3 | 729 | 4 | 21 | 15 | 18 | 15309 | 10935 | 26338 |
| 4 | 6561 | 8 | 28 | 20 | 24 | 183708 | 131220 | 322312 |

associated with a continuous distribution such as the normal distribution, regardless of the use of variance reduction techniques.

The number of iterations continues to be significantly greater in Model-I with respect to Model-II, therefore the time required to solve the problem is greater in Model-I as observed in Table 6.

From Table 6, as the problem increases in periods, the time taken to find the solution of Model-I is longer. In all cases, a global optimum was found for Model-II, for Model-I only a feasible solution was found in period 4. Moreover, an advantage can also be observed from the perspective of the number of iterations and CPU Time after 3 periods. number of iterations and CPU Time after 3 periods.

Fig 3 reports the Expected Value (EV) indicator, which is the expectation of all scenarios, Lingo reports this value also as the value of the objective function. The values are close (see Fig 2) and this causes a low gap between both models, so from a computational perspective the Model-II is convenient, even more if the interest is to seek quick decisions in organizations.

Fig 4 reports the wait-and-see (WS) indicator, this reports the expected value of the objective function by removing the nonanticipativity constraints. In practice, this is impossible due to the randomness cannot be anticipated. The values tend to be very similar; this is because the approximation using the maximum likelihood values of the normal distribution, minimize the error of the approximation.

Generally, the use of heuristic and computational techniques to approximate the solution is a motive for research, seeking speed with the cost of a solution. From Fig 3 it can be seen that from the perspective of the WS solution the proposed model (Model-II) closely approximates to the real problem unlike other studies (see [29, 45]).

Fig 5 reports the indicator perfect Information of expected value (EVPI) is the absolute value of the difference between EV and WS. It is the maximum amount that would be paid to obtain information [22] and reduce randomness. It should be mentioned that the existence of a high EVPI justifies the use of a stochastic model, rather than using a deterministic model. Table 7 summarizes what was mentioned about the EV, WS and EVPI stochastic optimization indicators, adding the EV GAP. Note that the GAP obtained is low, so both models are similar in their result of the objective value as already mentioned above.

**Table 6. Comparison between Model-I and Model-II regarding the time it takes to solve the problem and the number of iterations.** Own elaboration.

| Periods | Scenarios | R. V. | CPU Time (s) | | Iterations | | Type of solution found | |
|---|---|---|---|---|---|---|---|---|
| | | | Model-I | Model-II | Model-I | Model-II | Model-I | Model-II |
| 2 | 81 | 2 | 1.53 | 0.47 | 6958 | 9877 | Global optimal | Global optimal |
| 3 | 729 | 4 | 10.08 | 4.37 | 133406 | 96704 | Global optimal | Global optimal |
| 4 | 6561 | 8 | 7200* | 654.3 | 4736377 | 1406895 | Feasible solution | Global optimal |

"*" Indicates that the problem was stopped due to not finding a better solution (R.V. denotes random variables)

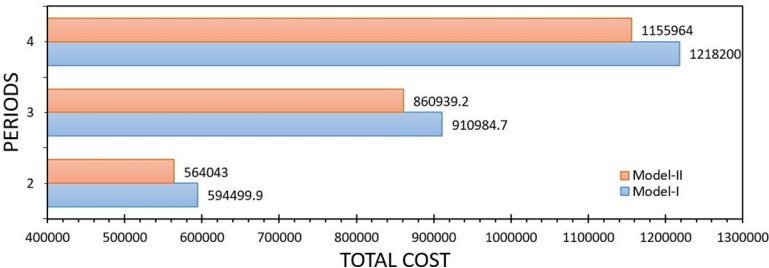

**Fig 3. Comparison between Model-I and Model-II with respect to the expected value solution.** Own elaboration.

These results show the efficiency of the proposal Model-II even when there are two random variables, it can be noted that when the random variables are associated with a normal distribution with known parameters and in a certain way compact (i.e. standard deviation and mean close), it allows the discretization and generation of the scenario tree to be efficient computationally.

When continuous distributions are approximated with discrete distributions, it can be observed that it affects the quality of the Model-II solution, despite this drawback, the quality of the solution is still acceptable in terms of the EV GAP, in this case about 5%.

## 6. Sensitivity analysis

In this section a sensitivity analysis is carried out to observe how robust the objective function is, and the decision variables are when costs are changing. The problem with 3 periods was taken as a reference for both models.

For Model-I, the middle scenario and the expected value of the objective function are reported. For Model-II, the most probable scenario and the expected value of the objective function are reported. The decision variables that were considered indicate in what period they are. Tables 7 and 8 show how the model change with a cost variation (in percentage), in addition to the fact that Model-I and Model-II are remarkably similar to each other with the results.

When performing this sensitivity analysis (Tables 8 and 9), it can be observed that there are parameters that greatly affect the objective function, such as production costs $C_p$ and product production costs $C_X$, if it varies towards positive and negative percentages, and the firing cost $C_F$, when it varies negatively, however, there is a small difference in terms of the number of workers assigned to production in the first period $P_1$ (first state decision).

In general, the variation of the objective function is similar between both models except for the variation in $C_p$ which implies a greater increase in Model-I with respect to Model-II, and

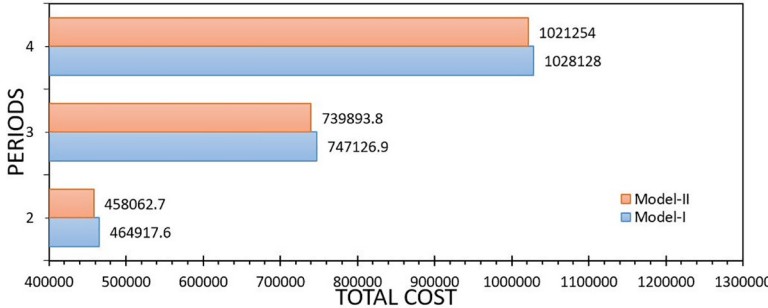

**Fig 4. Comparison between Model-I and Model-II with respect to the wait-and-see solution.** Own elaboration.

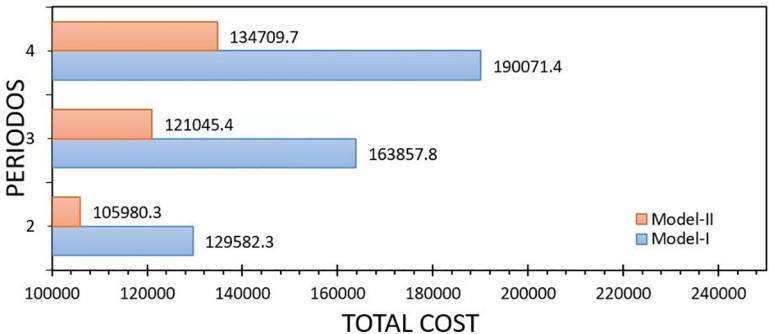

**Fig 5. Comparison between Model-I and Model-II with respect to the expected value for perfect information.**
Own elaboration.

in $C_F$ which results have greater reduction of Model-II with respect to Model-I. One consideration is the hiring policy, Model-I considers hiring in 11 of the 12 sensitivity results, this because it must be considered that the demand, from a statistical perspective, has infinite achievements. In order to deal with this, the resource variable evaluates how many workers to hire.

On the other hand, Model-II considers that there are only three possible realizations, therefore its evaluation from a computational perspective is less complex. As there are not infinite realizations, the corrective actions by the resource variables are minimal. Another consideration is when $C_F$ is null, it should be noted that the model prefers to hire and assign many workers $P_1$ and then carry out the fired workers, this suggests that an outsourcing policy would mean a viable strategy for the company in a situation where the behavior of demand is known.

For the cost $C_I$, $C_R$ and $C_S$ there were no large significant changes in the objective function, but reducing $C_R$, it is possible to have a smaller workforce in the first periods, to later carry out hiring without having an additional cost, thus indirectly reducing inventory costs and production.

It is clear, that the proposed Model-II turns out to be in general very similar as mentioned previously, perhaps it would be worthwhile to analyze in detail the hiring policies obtained through the variation of parameters, particularly in economic situations where companies see their capital reduced and they are forced to decrease payments or costs could increase.

## 6.1 Impact of service level on Model-I and Model-II

In this section, an analysis on the impact of the service level in Model-I and Model-II is developed, considering the same scenarios and number of periods previously used, starting from 86% to 98% of service level. Because the service level restriction is important in the model, the

**Table 7. Comparison between Model-I and Model-II with respect to the importance indicators of stochastic optimization, "*" indicates approximation.** Own elaboration.

| | EV | | WS | | EVPI | | EV GAP (%) |
|---|---|---|---|---|---|---|---|
| **Periods** | **Model-I** | **Model-II** | **Model-I** | **Model-II** | **Model-I** | **Model-II** | |
| **2** | 594499.9 | 564043 | 464917.6 | 458062.7 | 129582.3 | 105980.3 | 5.1231 |
| **3** | 910984.7 | 860939.2 | 747126.9 | 739893.8 | 163857.8 | 121045.4 | 5.4936 |
| **4** | 1218200 | 1155964 | 1028128 | 1021254 | 190071.4 | 134709.7 | 5.1088* |

**Table 8. Sensitivity analysis of Model-I.** Own elaboration.

| Case | Parameter | Variation | $P_1$ | $R_2$ | $F_2$ | $X_1$ | $I_1$ | $S_3$ | Δ-Cost (%) |
|------|-----------|-----------|-------|-------|-------|-------|-------|-------|-----------|
| Base case | - | - | 34 | 3 | 0 | 346 | 199 | 38 | - |
| 1 | $C_P$ | 50% | 34 | 3 | 0 | 349 | 202 | 38 | 78.0267 |
| 2 | | -50% | 34 | 3 | 0 | 346 | 199 | 38 | -38.618 |
| 3 | $C_F$ | 100% | 34 | 3 | 0 | 346 | 199 | 38 | 1.126 |
| 4 | | -100% | 50 | 0 | 23 | 515 | 368 | 38 | -16.950 |
| 5 | $C_R$ | 100% | 34 | 3 | 0 | 346 | 199 | 38 | 0.7928 |
| 6 | | -100% | 34 | 5 | 0 | 346 | 199 | 38 | -0.7928 |
| 7 | $C_I$ | 200% | 34 | 3 | 0 | 346 | 199 | 38 | 0.7694 |
| 8 | | -100% | 34 | 3 | 0 | 350 | 203 | 38 | -0.3849 |
| 9 | $C_5$ | 100% | 34 | 3 | 0 | 346 | 199 | 37 | 0.2488 |
| 10 | | -100% | 34 | 3 | 0 | 346 | 199 | 38 | -0.2489 |
| 11 | $C_X$ | 100% | 34 | 3 | 0 | 346 | 199 | 38 | 20.2578 |
| 12 | | -100% | 34 | 3 | 0 | 346 | 199 | 38 | -20.482 |

sensitivity should be analyzed in the same way as another variation parameters of the model. From Tables 10 and 11 it can be observed that, at a lower service level of Model-I and Model-II, the objective function is lower (Expected Value). The EV solution gradually increases as the service level increases.

For the indicator EVPI, this increase does not occur, in the case of Model-I a lower EVPI implies that the model is not so affected by random issues, therefore, managing a service level of 90% is viable for the Model- I. However, in Model-II we find that with a service level of 92%, the model does not show positive increases, hence, managing this level of service allows reducing the EVPI, but this company policy implies a higher cost due to the increment in the EV.

It can be noticed that the best service level policies are obtained with low service levels, which can improve EV and EVPI, but this would imply too many backlogs that could leave a not unfavorable impression of the company. The other option is to manage a service level policy of between 90% and 92%, being able to improve EVPI and without a drastic increase in EV,

**Table 9. Sensitivity analysis of Model-II.** Own elaboration.

| Case | Parameter | Variation | $P_1$ | $R_2$ | $F_2$ | $X_1$ | $I_1$ | $S_3$ | Δ-Cost (%) |
|------|-----------|-----------|-------|-------|-------|-------|-------|-------|-----------|
| Base case | - | - | 34 | 3 | 0 | 346 | 199 | 38 | - |
| 1 | $C_P$ | 50% | 34 | 3 | 0 | 349 | 202 | 38 | 78.0267 |
| 2 | | -50% | 34 | 3 | 0 | 346 | 199 | 38 | -38.618 |
| 3 | $C_F$ | 100% | 34 | 3 | 0 | 346 | 199 | 38 | 1.126 |
| 4 | | -100% | 50 | 0 | 23 | 515 | 368 | 38 | -16.950 |
| 5 | $C_R$ | 100% | 34 | 3 | 0 | 346 | 199 | 38 | 0.7928 |
| 6 | | -100% | 34 | 5 | 0 | 346 | 199 | 38 | -0.7928 |
| 7 | $C_I$ | 200% | 34 | 3 | 0 | 346 | 199 | 38 | 0.7694 |
| 8 | | -100% | 34 | 3 | 0 | 350 | 203 | 38 | -0.3849 |
| 9 | $C_S$ | 100% | 34 | 3 | 0 | 346 | 199 | 37 | 0.2488 |
| 10 | | -100% | 34 | 3 | 0 | 346 | 199 | 38 | -0.2489 |
| 11 | $C_X$ | 100% | 34 | 3 | 0 | 346 | 199 | 38 | 20.2578 |
| 12 | | -100% | 34 | 3 | 0 | 346 | 199 | 38 | -20.482 |

**Table 10. Impact of the service level constraint in Model-I.** Own elaboration.

| Service level (%) | Δ-EV (%) | EV | Δ-WS (%) | Δ-EVPI (%) |
|---|---|---|---|---|
| 86 | -0.01111753 | 900856.8 | -0.014417899 | 0.00393085 |
| 88 | -0.0059372 | 905576 | -0.007466469 | 0.00103565 |
| 90* | 0 | 910984.7 | 0 | 0 |
| 92 | 0.00886085 | 919056.8 | 0.007310539 | 0.01592967 |
| 94 | 0.01651784 | 926032.2 | 0.014569279 | 0.0254019 |
| 96 | 0.02247535 | 931459.4 | 0.022478243 | 0.02246216 |
| 98 | 0.0359522 | 943736.6 | 0.029948995 | 0.06332442 |

"*" indicates base case.

however, the cost is higher, even though the level of service is good, and that is easy to agree between company and clients. Tables 12 and 13 show that the level of service significantly affects the decision variables. It is evident that as the level of service increases, more workers are assigned to production, and with this measure the company produces more units. This also increases the inventory level; with these countermeasures the backlogs decrease.

## 6.2 Impact of variation in the distribution probability parameters of the random variables

Finally, an analysis on the impact of variation in the distribution probability in Model-I and Model-II is developed, considering the same scenarios and number of periods previously used in the las two sections, two different cases were analyzed, the first where the mean has different values, but with the same standard deviation and the second case, where the mean is the same, but the standard deviation has different values. Figs 6 and 7 explain what has been done.

Table 14 presents the sensitivity for the normal distribution parameters (mean and standard deviation) of the two random variables (production capacity and demand). Cases 1 to 3 and 7 to 9 show how the behavior is when the mean is fixed, and the standard deviation varies for both random variables.

Cases 4 to 6 and 10 to 12 show the sensitivity achieved when the mean of both probability distributions is varied, leaving the standard deviation corresponding to each random variable fixed. It is evident that the best results regarding EV are obtained in cases 1, 6, 9 and 10, which are when the standard deviation decreases for production capacity or its mean increases, as well as when there is a high variation in demand or its average low.

**Table 11. Impact of the service level constraint in Model-II.** Own elaboration.

| Service level (%) | Δ-EV (%) | EV | Δ-WS (%) | Δ-EVPI (%) |
|---|---|---|---|---|
| 86 | -0.01576139 | 847369.6 | -0.01356681 | -0.029175 |
| 88 | -0.01051677 | 851884.9 | -0.0071567 | -0.03105529 |
| 90* | 0 | 860939.2 | 0 | 0 |
| 92 | 0.00534684 | 865542.5 | 0.00797547 | -0.01072077 |
| 94 | 0.02064327 | 878711.8 | 0.01516109 | 0.05415324 |
| 96 | 0.02593551 | 883268.1 | 0.02204208 | 0.04973423 |
| 98 | 0.02876963 | 885708.1 | 0.02942476 | 0.02476509 |

"*" indicates base case

**Table 12. Impact of the service level constraint respect to decision variables in Model-I.** Own elaboration.

| Service level (%) | $P_1$ | $R_1$ | $F_2$ | $X_1$ | $I_1$ | $S_3$ |
|---|---|---|---|---|---|---|
| 86 | 33 | 0 | 1 | 336 | 189 | 54 |
| 90 | 34 | 0 | 0 | 346 | 199 | 38 |
| 94 | 35 | 0 | 0 | 360 | 213 | 23 |
| 98 | 36 | 0 | 0 | 366 | 219 | 7 |

For the decision variables, the greatest changes occur when the parameters of the random variable demand are varied, in case 12, particularly for the variables $P_1$, $S_3$ y $X_1$. In the case of the random variable production capacity, its behavior is like that found in Model-I, that a lower variation in production capacity or that its average is higher leads to better results.

Comparing both random variables, varying the standard deviation of the production capacity has more impact on the decision variables and on the model in general, but if the mean is varied, the demand has more impact, this because the cost of the production plan will be proportional to the demand.

Table 15 is the counterpart of Table 14, where the parameters of the discrete distributions (their values and probabilities of occurrence) were obtained using the Gaussian quadrature method proposed in the methodology. It can be observed in Table 15 similar behaviors in terms of the variation of the parameters of the probability distribution for production capacity, than those observed for Model-I.

It should be noted that the best results with respect to the EV indicator occur again in cases 1, 6, 10 and now in case 7. For case 10 the changes in the EV indicator They are explained because less products are produced and the inventory reduces compared to the base case, this is explained by the decrease in demand. Again, the higher the demand, it is clearly observed in the results how the decision variables are affected. These results show the quality of Model-II, using the proposed methodology because the solutions are similar.

## 7. Conclusions

This research considers an aggregate production plan applied to a company. Aggregate production plans play an important role in SMEs because they allow them to manage their resources and operations efficiently. This plays an important role in planning as it reduces the risk of SMEs disappearing, particularly in Mexico. Due to various policies such as hiring and firing, the production capacity in the company was not constant, which is why it was initially considered as a random variable, later demand is incorporated as a second random variable. Aggregate production plans under uncertainty have been studied in the literature for different types of problem structures such as linear optimization models, mixed linear-integers, non-linear-mixed integers, among others, in addition to whether they are single-objective or multiple-objective. Although some studies report solutions under the here and now or wait and see

**Table 13. Impact of the service level constraint respect to decision variables in Model-II.** Own elaboration.

| Service level (%) | $P_1$ | $R_1$ | $F_2$ | $X_1$ | $I_1$ | $S_3$ |
|---|---|---|---|---|---|---|
| 86 | 31 | 0 | 1 | 364 | 211 | 49 |
| 90 | 32 | 0 | 2 | 381 | 228 | 35 |
| 94 | 33 | 0 | 3 | 395 | 242 | 21 |
| 98 | 34 | 0 | 2 | 390 | 237 | 7 |

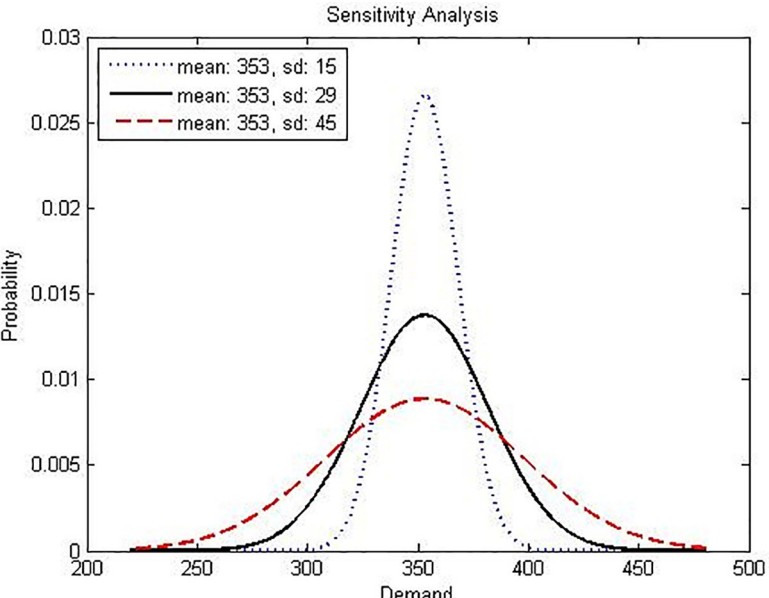

**Fig 6. Graphical interpretation of the sensitivity analysis for Model-I varying standard deviation parameter for the random variable demand.** Own elaboration.

solution, both solutions have never been considered in the same study, in addition we incorporated the result of the expected value of perfect information in this work.

Model-I could not solve the problem for more than three periods, for which a second model was proposed (Model-II), following a proposed methodology, achieving good results when comparing both models that is, there is no significant difference between the objective

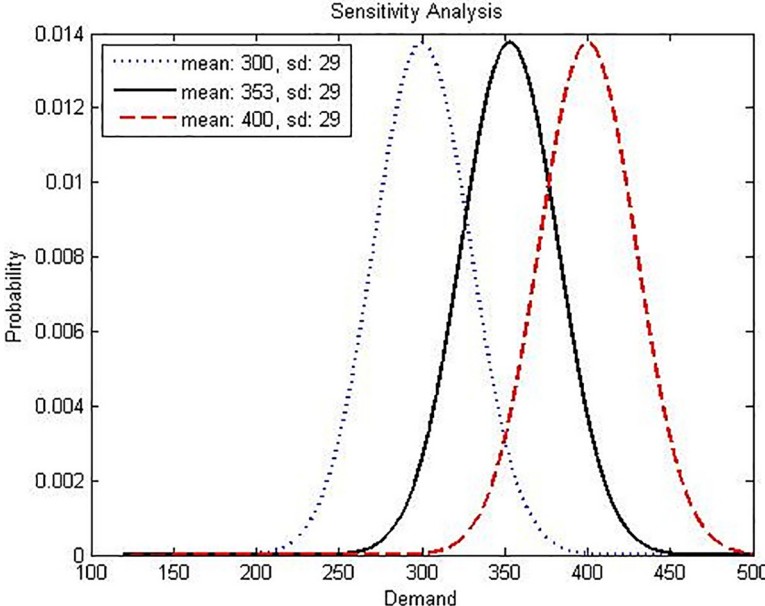

**Fig 7. Graphical interpretation of the sensitivity analysis for Model-I varying mean parameter for the random variable demand.** Own elaboration.

**Table 14. Model-I sensitivity analysis varying parameters of the probability distribution.** Own elaboration.

| Case number | Parameters | | | | Decision variables | | | | | Δ−EV (%) | Δ−WS (%) | Δ−EVPI (%) |
|---|---|---|---|---|---|---|---|---|---|---|---|---|
| | Random variable | $\mu$ | $\sigma$ | $P_1$ | $R_1$ | $F_2$ | $X_1$ | $I_1$ | $S_3$ | | | |
| *Variation of the standard deviation σ of the production capacity* | | | | | | | | | | | | |
| 1 | | 12 | 1 | 28 | 0 | 0 | 310 | 163 | 38 | -13.184 | -1.168 | -67.972 |
| 2* | $k_t^\omega$ | 12 | 2 | 34 | 0 | 0 | 346 | 199 | 38 | 0.000 | 0.000 | 0.000 |
| 3 | | 12 | 3 | 46 | 0 | 2 | 433 | 286 | 38 | 23.583 | 2.776 | 118.457 |
| *Variation of the mean μ of the production capaicty* | | | | | | | | | | | | |
| 4 | | 10 | 2 | 45 | 0 | 0 | 369 | 222 | 38 | 25.065 | 15.731 | 67.621 |
| 5* | | 12 | 2 | 34 | 0 | 0 | 346 | 199 | 38 | 0.000 | 0.000 | 0.000 |
| 6 | | 14 | 2 | 27 | 0 | 0 | 325 | 178 | 38 | -15.005 | -10.917 | -33.643 |
| *Variation of the standard deviation σ of the demand* | | | | | | | | | | | | |
| 7 | | 353 | 15 | 35 | 0 | 0 | 357 | 207 | 37 | 2.821 | 0.189 | 14.819 |
| 8* | $D_t^\delta$ | 353 | 29 | 34 | 0 | 0 | 346 | 199 | 38 | 0.000 | 0.000 | 0.000 |
| 9 | | 353 | 45 | 32 | 1 | 0 | 326 | 182 | 40 | -2.948 | -0.055 | -16.138 |
| *Variation of the mean μ of the demand* | | | | | | | | | | | | |
| 10 | | 300 | 29 | 28 | 0 | 0 | 288 | 194 | 33 | -17.409 | -16.589 | -21.148 |
| 11* | | 353 | 29 | 34 | 0 | 0 | 346 | 199 | 38 | 0.000 | 0.000 | 0.000 |
| 12 | | 400 | 29 | 40 | 1 | 0 | 411 | 217 | 43 | 15.863 | 14.761 | 20.887 |

"*" denotes the base case.

function, with model 2 being the most effective at the computational level. The advantages offered by the methodology proposed is its flexibility, being able to use it in other problems where there is uncertainty. Some of its limitations, are the need for goodness of fit tests that ensure that the data have a certain probability distribution; if they are not done, the results will be far from optimal. Also, the sample size or number of possible realizations of a random variable when it is discretized can make the problem difficult to solve, because the equivalent

**Table 15. Model-II sensitivity analysis varying parameters of the probability distribution.** Own elaboration, "*" denotes the base case.

| Case | Parameters | | Decision variables | | | | | | Δ−EV (%) | Δ−WS (%) | Δ−EVPI (%) |
|---|---|---|---|---|---|---|---|---|---|---|---|
| | RV*. | Values and probabilities | $P_1$ | $R_1$ | $F_2$ | $X_1$ | $I_1$ | $S_3$ | | | |
| *Approximation to the variation of the standard deviation σ of the capacity production* | | | | | | | | | | | |
| 1 | | $k_1 = 11\ k_2 = 12\ k_3 = 13$ $P(k_1) = 0.25$ $P(k_2) = 0.5$ $P(k_3) = 0.25$ | 29 | 0 | 0 | 341 | 188 | 35 | -6.216 | -0.504 | -41.133 |
| 2* | $k_t^\omega$ | $k_1 = 10\ k_2 = 12\ k_3 = 14$ $P(k_1) = 0.267$ $P(k_2) = 0.466$ $P(k_3) = 0.267$ | 32 | 0 | 2 | 381 | 228 | 35 | 0.000 | 0.000 | 0.000 |
| 3 | | $k_1 = 9\ k_2 = 12\ k_3 = 15$ $P(k_1) = 0.3$ $P(k_2) = 0.4$ $P(k_3) = 0.3$ | 35 | 0 | 3 | 403 | 250 | 35 | 7.160 | 0.993 | 44.862 |

*(Continued)*

**Table 15.** (Continued)

| Case | Parameters | | Decision variables | | | | | | Δ−EV (%) | Δ−WS (%) | Δ−EVPI (%) |
|---|---|---|---|---|---|---|---|---|---|---|---|
| | RV*. | Values and probabilities | $P_1$ | $R_1$ | $F_2$ | $X_1$ | $I_1$ | $S_3$ | | | |
| *Approximation to the variation of the mean μ of the capacity production* | | | | | | | | | | | |
| 4 | | $k_1 = 8$ $k_2 = 10$ $k_3 = 12$ | 40 | 0 | 3 | 387 | 234 | 35 | 19.084 | 14.953 | 44.332 |
| | | $P(k_1) = 0.267$ | | | | | | | | | |
| | | $P(k_2) = 0.466$ | | | | | | | | | |
| | | $P(k_3) = 0.267$ | | | | | | | | | |
| 5* | | $k_1 = 10$ $k_2 = 12$ $k_3 = 14$ | 32 | 0 | 2 | 381 | 228 | 35 | 0.000 | 0.000 | 0.000 |
| | | $P(k_1) = 0.267$ | | | | | | | | | |
| | | $P(k_2) = 0.466$ | | | | | | | | | |
| | | $P(k_3) = 0.267$ | | | | | | | | | |
| 6 | | $k_1 = 12$ $k_2 = 14$ $k_3 = 16$ | 27 | 0 | 1 | 355 | 202 | 35 | -12.347 | -10.69 | -22.503 |
| | | $P(k_1) = 0.267$ | | | | | | | | | |
| | | $P(k_2) = 0.466$ | | | | | | | | | |
| | | $P(k_3) = 0.267$ | | | | | | | | | |
| *Approximation to the variation of the standard deviation σ of the demand* | | | | | | | | | | | |
| 7 | | $D_1 = 338$ $D_2 = 353$ $D_3 = 368$ | 30 | 0 | 0 | 353 | 200 | 35 | -2.857 | -0.02 | -20.194 |
| | | $P(D_1) = 0.225$ | | | | | | | | | |
| | | $P(D_2) = 0.55$ | | | | | | | | | |
| | $D_t^\delta$ | $P(D_3) = 0.225$ | | | | | | | | | |
| 8* | | $D_1 = 324$ $D_2 = 353$ $D_3 = 382$ | 32 | 0 | 2 | 381 | 228 | 35 | 0.000 | 0.000 | 0.000 |
| | | $P(D_1) = 0.25$ | | | | | | | | | |
| | | $P(D_2) = 0.5$ | | | | | | | | | |
| | | $P(D_3) = 0.25$ | | | | | | | | | |
| 9 | | $D_1 = 308$ $D_2 = 353$ $D_3 = 398$ | 33 | 0 | 2 | 390 | 237 | 35 | 2.561 | 0.166 | 17.198 |
| | | $P(k_1) = 0.3$ | | | | | | | | | |
| | | $P(k_2) = 0.4$ | | | | | | | | | |
| | | $P(k_3) = 0.3$ | | | | | | | | | |
| *Approximation to the variation of the mean μ of the demand* | | | | | | | | | | | |
| 10 | | $D_1 = 271$ $D_2 = 300$ $D_3 = 329$ | 27 | 0 | 1 | 306 | 137 | 30 | -15.614 | -16.47 | -10.381 |
| | | $P(k_1) = 0.25$ | | | | | | | | | |
| | | $P(k_2) = 0.5$ | | | | | | | | | |
| | | $P(k_3) = 0.25$ | | | | | | | | | |
| 11* | | $D_1 = 324$ $D_2 = 353$ $D_3 = 382$ | 32 | 0 | 2 | 381 | 228 | 35 | 0.000 | 0.000 | 0.000 |
| | | $P(D_1) = 0.25$ | | | | | | | | | |
| | | $P(D_2) = 0.5$ | | | | | | | | | |
| | | $P(D_3) = 0.25$ | | | | | | | | | |
| 12 | | $D_1 = 371$ $D_2 = 400$ $D_3 = 429$ | 36 | 1 | 0 | 416 | 216 | 40 | 13.748 | 14.655 | 8.209 |
| | | $P(k_1) = 0.25$ | | | | | | | | | |
| | | $P(k_2) = 0.5$ | | | | | | | | | |
| | | $P(k_3) = 0.25$ | | | | | | | | | |

*RV is Random Variables

deterministic grow exponentially as the number of states increases, for this study; solving a fifth state would imply solving a deterministic equivalent of more than two million decision variables and four million constraints because more than 40,000 scenarios are required, which

is the maximum number of scenarios that the Lingo software can process without having a computational memory deficit.

The study was complemented with sensitivity analysis, in the literature few studies report these analyzes, generally only perform it by varying parameters associated with costs, but in this study is carried out to see the impact of varying the percentage of the policy of level of service, also, a sensitivity analysis is also carried out, varying the parameters of the probability distributions or stochastic parameters (mean and standard deviation) and evaluate the impacts in the solutions and decision variables, so that the company has sufficient information for correct planning in case these parameters could change in the future, or, as an area of opportunity to improve productivity, for example, it could be observed that reducing the variability of the random variable production capacity, that is, reducing its standard deviation, reduces the total cost of the APP.

Since many of the APPs occupy neither linear functions, a direction for future research is to make a model considering some non-linear functions, such as the inventory cost, also reformulate the problem, removing some restrictions that will allow to have a lower inventory level, allowing to solve the problem in a more efficient way, it is also interested in the use of another algorithm to solve the equivalent determinists, in this work the algorithm B-and-B was used, being able also to use algorithms of cut of plan, consider using multiple kernels in parallel, using multiple heuristics to pre-solve the problem, and using robust algorithms for relaxation of the problem.

## Author Contributions

**Conceptualization:** José Emmanuel Gómez-Rocha, Eva Selene Hernández-Gress, Héctor Rivera-Gómez.

**Data curation:** José Emmanuel Gómez-Rocha.

**Formal analysis:** Eva Selene Hernández-Gress.

**Investigation:** José Emmanuel Gómez-Rocha, Héctor Rivera-Gómez.

**Methodology:** José Emmanuel Gómez-Rocha, Eva Selene Hernández-Gress, Héctor Rivera-Gómez.

**Software:** José Emmanuel Gómez-Rocha.

**Supervision:** Eva Selene Hernández-Gress, Héctor Rivera-Gómez.

**Validation:** José Emmanuel Gómez-Rocha, Héctor Rivera-Gómez.

**Writing – original draft:** José Emmanuel Gómez-Rocha, Eva Selene Hernández-Gress, Héctor Rivera-Gómez.

**Writing – review & editing:** Eva Selene Hernández-Gress.

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
