## [Decision Letter · Decision Letter 0]

17 Mar 2021

PONE-D-21-01331

Production planning of a furniture manufacturing company with random demand and production capacity using stochastic programming

PLOS ONE

Dear Dr. Hernández Gress,

Thank you for submitting your manuscript to PLOS ONE. After careful consideration, we feel that it has merit but does not fully meet PLOS ONE’s publication criteria as it currently stands. Therefore, we invite you to submit a revised version of the manuscript that addresses the points raised during the review process.

We look forward to receiving your revised manuscript.

Kind regards,

Dragan Pamucar

Academic Editor

PLOS ONE

Journal Requirements:

"We appreciate the support of the company to carry out this research."

Additionally, because some of your funding information pertains to commercial funding, we ask you to provide an updated Competing Interests statement, declaring all sources of commercial funding.

In your Competing Interests statement, please confirm that your commercial funding does not alter your adherence to PLOS ONE Editorial policies and criteria by including the following statement: "This does not alter our adherence to PLOS ONE policies on sharing data and materials.” as detailed online in our guide for authors  http://journals.plos.org/plosone/s/competing-interests.  If this statement is not true and your adherence to PLOS policies on sharing data and materials is altered, please explain how.

Please include the updated Competing Interests Statement and Funding Statement in your cover letter. We will change the online submission form on your behalf.

Reviewers' comments:

Reviewer's Responses to Questions

**Comments to the Author**

1. Is the manuscript technically sound, and do the data support the conclusions?

Reviewer #1: Yes

Reviewer #2: No

2. Has the statistical analysis been performed appropriately and rigorously? 

Reviewer #1: Yes

Reviewer #2: No

3. Have the authors made all data underlying the findings in their manuscript fully available?

Reviewer #1: Yes

Reviewer #2: Yes

4. Is the manuscript presented in an intelligible fashion and written in standard English?

Reviewer #1: Yes

Reviewer #2: No

5. Review Comments to the Author

Reviewer #1: Thank you for inviting me as a reviewer for the paper titled Production planning of a furniture manufacturing company with random demand and production capacity using stochastic programming. The paper has well structure. The strengths of this paper are an interesting topic and good methodology.

However, the authors need to consider the following minor/major points as a limitation or further scope for refining the paper:

- Clearly define motivations for this study.

- Need to highlight the novelty of the study in the introduction.

- The Methodology section adds a figure, which describes the stages and steps in applying the presented methodology.

- Literature analysis needs to be improved. In the entire paper, there is only one cited paper from 2019, while from 2020 and 2021 there are no papers. Add another 10-15 papers of more recent date (period 2019-2021), such as:

o Giri, B. C., & Dey, S. (2020). Game theoretic models for a closed-loop supply chain with stochastic demand and backup supplier under dual channel recycling. https://doi.org/10.31181/dmame2003015g;

o Çam, Ömer N., & Sezen, H. K. (2020). The formulation of a linear programming model for the vehicle routing problem in order to minimize idle time. https://doi.org/10.31181/dmame2003132h

-Add advantages and limitations of the proposed methodology and this study.

Reviewer #2: This work proposes two multi-stage stochastic linear programming models, which are applied to an aggregate production plan (APP) for a furniture manufacturing company located in the state of Hidalgo, Mexico. I have read the manuscript with great interest and found that the idea is interesting but presentation style is poor. From my point of view, its quality and novelty is not enough to be published in PLOS ONE journal. I try to justify my opinion:

Language of the paper must be improved as there are many mistakes and it is hard to clearly understand the flow of the paper.

Objectives of this work are not convincing.

The novelty of this work is not clear.

Abstract and Introduction section have poorly written. Introduction section should be updated with motivation behind this study, research gap, novelty and contributions of the work.

Literature review is just providing summary of existing works without any insightful conclusions.

The structure of the whole paper should be improved.

Why someone should use your proposed work in practice, and what are the advantages of your work in comparison with others.

Summarise the advantages and limitations of the proposed method in practical applications.

Comparison with existing studies should be added.

Conclusions section has poorly written. Conclusion section should be rewritten by adding advantages, limitations and useful future research directions of the proposed work.

References are not consistent.

6. PLOS authors have the option to publish the peer review history of their article (what does this mean?). If published, this will include your full peer review and any attached files.

Reviewer #1: No

Reviewer #2: No

---

## [Author Response · Author response to Decision Letter 0]

2 May 2021

Dear reviewers:

We appreciate the time you have invested in reviewing our article. We have tried to attend to the observations, below we respond specifically to each one of them.

Edditor comments: 

 We attend the style requirements, including those for file naming.

"We appreciate the support of the company to carry out this research."

Additionally, because some of your funding information pertains to commercial funding, we ask you to provide an updated Competing Interests statement, declaring all sources of commercial funding.

In your Competing Interests statement, please confirm that your commercial funding does not alter your adherence to PLOS ONE Editorial policies and criteria by including the following statement: "This does not alter our adherence to PLOS ONE policies on sharing data and materials.” as detailed online in our guide for authors http://journals.plos.org/plosone/s/competing-interests. If this statement is not true and your adherence to PLOS policies on sharing data and materials is altered, please explain how. Please include the updated Competing Interests Statement and Funding Statement in your cover letter. We will change the online submission form on your behalf.

We don´t have any funding for the research, only the company provided us the data. In order to attend this comment, we decided to disappear Acknowledgments of the article and using the Funding Stament as "The authors have declared that no competing interests exist."

We review the competing interest, and we do not have any of them.

Reviewers' comments:

Reviewer's Responses to Questions

Comments to the Author

1. Is the manuscript technically sound, and do the data support the conclusions?

Reviewer #1: Yes

Reviewer #2: No

We include the data and all of the statistical analysis in this link https://doi.org/10.6084/m9.figshare.14444609.v1 . Also we include the lingo models in https://doi.org/10.6084/m9.figshare.14450430. Both the data and the lingo models could be used to replicate the results.

2. Has the statistical analysis been performed appropriately and rigorously? 

Reviewer #1: Yes

Reviewer #2: No

We include the data and all of the statistical analysis in this link https://doi.org/10.6084/m9.figshare.14444609.v1 , and could be used to replicate the results 

3. Have the authors made all data underlying the findings in their manuscript fully available?

Reviewer #1: Yes

Reviewer #2: Yes

We include the data and all of the statistical analysis in this link https://doi.org/10.6084/m9.figshare.14444609.v1 . Also we include the lingo models in https://doi.org/10.6084/m9.figshare.14450430. Both the data and the lingo models could be used to replicate the results.

4. Is the manuscript presented in an intelligible fashion and written in standard English?

Reviewer #1: Yes

Reviewer #2: No

We review our article with a native American speaker and we correct the typographical o and grammatical errors

5. Review Comments to the Author

Reviewer #1: Thank you for inviting me as a reviewer for the paper titled Production planning of a furniture manufacturing company with random demand and production capacity using stochastic programming. The paper has well structure. The strengths of this paper are an interesting topic and good methodology.

However, the authors need to consider the following minor/major points as a limitation or further scope for refining the paper:

- Clearly define motivations for this study.

The motivation is defined in the first paragraph of the introduction “This work presents an aggregate plan that was made for a company that manufactures furniture in the State of Hidalgo. Initially, a first approach to the solution of the problem was made in [1]. In this work, only the production capacity was considered as a random variable using two models, one with a continuous probability distribution and the other with a discrete one. However, another extremely important random variable had been ignored due to complexity: demand. Therefore, the motivation for this work is to improve productivity, have efficient policies to manage its production and minimize production costs, developing models of aggregate production plans (APP) with uncertainty due to a real need of a furniture company, Models are considering real characteristics such as human factor, multi-period production criteria and service level policy due to the use of backlogs”

- Need to highlight the novelty of the study in the introduction.

The novelty is in the third and fourth paragraphs : 

“The novelty of this work could be summarized in five points,1) this study provides a mathematical programming model that has been adapted for real needs of a company, which incorporates a service level constraint that it is not found in the literature, usually a confidence percentage is used (which could turn the problem into a chance constraint programming. 2) In the literature only the expected value of the objective function is reported (here and now solution) with the history of the process considered, that is, with the nonanticipativity constraints or the value of the expected objective function. If these constraints are removed, the wait and see solution appears, in our research, both solutions are reported, also the absolute difference of the two solutions is reported, called the expected value of perfect information that could help the company to deal with uncertainty in economic decisions. 3) An extensive sensitivity analysis is carried out, varying the cost parameters, the percentage of the service level and varying the parameters of the probability distribution of uncertainty. Few studies carry out a sensitivity analysis, but in our knowledge, nobody analyzes the impact of service level and varies the parameters of the probability distribution. Through this sensitivity analysis, interesting results were obtained, for example, that the total cost of the APP goes down, when the variability of the production capacity (standard deviation of the probability distribution) is reduced. 4) Due to the complexity of the problem, the software could not solve the problem satisfactorily for a fourth period, finding a solution that is only feasible, then, a second model is developed using discretization of the probability distribution, it has been shown that if the distances of both distributions are minimal, the solution found is closer than the true optimum the quality of the proposed model is presented in the results, where both models are compared.

Finally, 5) a methodology to deal with problems using stochastic programming is proposed, although it was applied to the case of this APP, can be implemented in other areas of industrial engineering sciences, such as supply chain networks, problems of vehicle routing, design, and redesign of layouts, among others. Advantages and disadvantages are detailed in the conclusions section. Table 1 shows a comparison between some relevant studies in the area and our study, so that the novelty and contribution of our proposal can be observed.”

- The Methodology section adds a figure, which describes the stages and steps in applying the presented methodology.

We improved this figure (Fig 2: Flow Chart of the Methodology)

- Literature analysis needs to be improved. In the entire paper, there is only one cited paper from 2019, while from 2020 and 2021 there are no papers. Add another 10-15 papers of more recent date (period 2019-2021), such as:

o Giri, B. C., & Dey, S. (2020). Game theoretic models for a closed-loop supply chain with stochastic demand and backup supplier under dual channel recycling. https://doi.org/10.31181/dmame2003015g;

o Çam, Ömer N., & Sezen, H. K. (2020). The formulation of a linear programming model for the vehicle routing problem in order to minimize idle time. https://doi.org/10.31181/dmame2003132h

We improve all the literature analysis, we add 12 papers of more recent period in the first three paragraphs of the literature review “Considering uncertainty within optimization problems remains a trending topic to be investigated because organizations face to stochastic variables when making decisions. A search was carried out in Scopus and in Web of Science written during 2020 and 2021 that used stochastic programming to solve optimization problems. Huang et. al [5] develop a multistage stochastic optimization model for system operators to efficiently schedule power-generation assets to co-optimize power generation and regulation reserve service under uncertainty. Ghayour et al. [6] present an approach called MLPR with linear programming used as its core in order to solve the influence maximization problem in the linear threshold model, that is one of two classic stochastic propagation models that describe the spread of influence in a network. Robust Multi-product Newsvendor Model with Substitution, where the demand and the substitution rates are stochastic and are subject to cardinality-constrained uncertainty sets that is an NP hard problem is presented in [7].

Also, Basciftci et. al [8] reformulate the robust facility location problem, in which they interpret the moments of stochastic demand as functions of facility-location decisions. In Shone et. al [9], stochastic modeling applications within aviation are presented, with a particular focus on problems involving demand and capacity management and the mitigation of air traffic congestion; using operations research perspective, including analytical queueing theory, stochastic optimal control, robust optimization and stochastic integer programming. Ghasemi et. al [10] present an Evolutionary Learning Based Simulation Optimization (ELBSO) method embedded within Ordinal Optimization. In ELBSO a Machine Learning (ML) based simulation metamodel is created using Genetic Programming (GP) to replace simulation experiments aimed at reducing computation; ELBSO is evaluated on a Stochastic Job Shop Scheduling Problem (SJSSP). Zhang et. al [11], consider a stochastic vehicle routing problem with probability constraints; the probability that customers are served before their (uncertain) deadlines must be higher than a pre-specified target. Wang et. al [12] propose a model to solve a project scheduling problem where resource assignments and activity schedules need to be determined to achieve a set of due-date requirements as well as possible. Torres et. al [13] present multistage stochastic program for the design and management of flexible infrastructure networks with stochastic demands.

 In the methods for solving stochastic programming, Dowson and Kapelevich [14] develop the Julia package for multistage stochastic and dual programming and Gangammanavar et. al [15] work with stochastic decomposition for two-stage stochastic linear programs with random cost coefficients.

Also we include a comparative table of the related works in Table 1 and an analysis of this table in the last paragraph “Reviewing the works found that are related to stochastic programming, it was observed that all of them use mixed integer programming, Jamalnia et al. [16], Zhao et al. [18] and Tirkolaee et al. [22] with nonlinear multiobjective, while Kazemi et al. [24] uses multistage stochastic programming which is the same as that used in our proposal. Tirkolaee et al. [22] use demand and costs as stochastic variables, while Kazemi et al. [24] demand and yield; the others, only use a single variable as a stochastic. In their approaches, a single approximation is used to explain the uncertainty, which is discrete, that is, a single stochastic model, and in our case two models, that allow us to compare between the normal distribution and its discretization in order to offer to the company a good solution in a reasonable computational time. The proposals found use the scenerio tree, except Tirkolaee et al. [22] that deal with the problem with weighted goal using GAMS. The level of service is only handled in Zhao et al. [18], but the impact of the service level on the solution is not considered. The sensitivity analysis varying stochastic parameters was not used in the approaches found. After analyzing the characteristics of the studies found, the gap was in considering the level of service, which was a very important restriction for the company in the case study, considering demand and labor as stochastic variables, which were two variables that generate a lot of uncertainty in the company and varying the stochastic parameters. Finally, generate an efficient model, that is to say, a model that obtains a good response in a reasonable computational time, for that reason 2 models were tested for comparison. 

-Add advantages and limitations of the proposed methodology and this study.

The advantages and limitations are in the second paragraph of the conclusions “The advantages offered by the methodology proposed is its flexibility, being able to use it in other problems where there is uncertainty. Some of its limitations, are the need for goodness of fit tests that ensure that the data have a certain probability distribution; if they are not done, the results will be far from optimal Also, the sample size or number of possible realizations of a random variable when it is discretized can make the problem difficult to solve, because the equivalent deterministic grow exponentially as the number of states increases, for this study; solving a fifth state would imply solving a deterministic equivalent of more than two million decision variables and four million constraints because more than 40,000 scenarios are required, which is the maximum number of scenarios that the Lingo software can process without having a computational memory deficit.”

Reviewer #2: This work proposes two multi-stage stochastic linear programming models, which are applied to an aggregate production plan (APP) for a furniture manufacturing company located in the state of Hidalgo, Mexico. I have read the manuscript with great interest and found that the idea is interesting but presentation style is poor. From my point of view, its quality and novelty is not enough to be published in PLOS ONE journal. I try to justify my opinion:

Language of the paper must be improved as there are many mistakes and it is hard to clearly understand the flow of the paper.

We review our article with a native American speaker and we correct the typographical o and grammatical errors

Objectives of this work are not convincing.

We modified the objectives of this work in the second paragraph of the introduction “The main objective of this article is to develop a multi-state stochastic optimization model applied to an APP of a local company, where the production periods are defined as the states, the randomness of production capacity and demand are modeled through a continuous probability distribution using the stochastic programming solver integrated by Lingo. Two models are proposed, Model-I only could solve the problem for a maximum of three periods, due the complexity of using a continuous probability distribution , a second model is proposed with a discretization of the probability distributions (Model-II) which could solve the problem up to four periods. In both models a scenario tree is created. In general, this work compares the efficiency between Model-I and Model-II in resolution time, number of iterations, expected value (EV), wait-and-see value (WS), and expected value of perfect information (EVPI). The obtained results help to determine the advantages about the proposed model (Model-II) with respect to Model I and is useful to understand the scope of both models and in which cases it is advisable to use each one. In addition, both models consider the impact of the service level restriction on the optimal solution and what happen when parameters of the distribution probabilities are varying”

The novelty of this work is not clear.

The novelty of this work could be summarized in five points and are in the third and fourth paragraph of the introduction “1) this study provides a mathematical programming model that has been adapted for real needs of a company, which incorporates a service level constraint that it is not found in the literature, usually a confidence percentage is used (which could turn the problem into a chance constraint programming. 2) In the literature only the expected value of the objective function is reported (here and now solution) with the history of the process considered, that is, with the nonanticipativity constraints or the value of the expected objective function. If these constraints are removed, the wait and see solution appears, in our research, both solutions are reported, also the absolute difference of the two solutions is reported, called the expected value of perfect information that could help the company to deal with uncertainty in economic decisions. 3) An extensive sensitivity analysis is carried out, varying the cost parameters, the percentage of the service level and varying the parameters of the probability distribution of uncertainty. Few studies carry out a sensitivity analysis, but in our knowledge, nobody analyzes the impact of service level and varies the parameters of the probability distribution. Through this sensitivity analysis, interesting results were obtained, for example, that the total cost of the APP goes down, when the variability of the production capacity (standard deviation of the probability distribution) is reduced. 4) Due to the complexity of the problem, the software could not solve the problem satisfactorily for a fourth period, finding a solution that is only feasible, then, a second model is developed using discretization of the probability distribution, it has been shown that if the distances of both distributions are minimal, the solution found is closer than the true optimum [2-4] the quality of the proposed model is presented in the results, where both models are compared.

Finally, 5) a methodology to deal with problems using stochastic programming is proposed, although it was applied to the case of this APP, can be implemented in other areas of industrial engineering sciences, such as supply chain networks, problems of vehicle routing, design, and redesign of layouts, among others. Advantages and disadvantages are detailed in the conclusions section. Table 1 shows a comparison between some relevant studies in the area and our study, so that the novelty and contribution of our proposal can be observed” 

Abstract and Introduction section have poorly written. Introduction section should be updated with motivation behind this study, research gap, novelty and contributions of the work.

The abstract was modified writing the novelty of our work in the last sentence “In this article two multi-stage stochastic linear programming models are developed, one applying the stochastic programming solver integrated by Lingo 17.0 optimization software that utilizes an approximation using an identical conditional sampling and Latin-hyper-square techniques to reduce the sample variance, associating the probability distributions to normal distributions with defined mean and standard deviation; and a second proposed model with a discrete distribution with 3 values and their respective probabilities of occurrence. In both cases, a scenario tree is generated. The models developed are applied to an aggregate production plan (APP) for a furniture manufacturing company located in the state of Hidalgo, Mexico, which has important clients throughout the country. Production capacity and demand are defined as random variables of the model. The main purpose of this research is to determine a feasible solution to the aggregate production plan in a reasonable computational time. The developed models were compared and analyzed. Moreover, this work was complemented with a sensitivity analysis; varying the percentage of service level, also, varying the stochastic parameters (mean and standard deviation) to test how these variations impact in the solution and decision variables”.

The introduction section was updated with research gag, novelty and contributions of the work “Therefore, the motivation for this work is to improve productivity, have efficient policies to manage its production and minimize production costs, developing models of aggregate production plans (APP) with uncertainty due to a real need of a furniture company, Models are considering real characteristics such as human factor, multi-period production criteria and service level policy due to the use of backlogs.

The main objective of this article is to develop a multi-state stochastic optimization model applied to an APP of a local company, where the production periods are defined as the states, the randomness of production capacity and demand are modeled through a continuous probability distribution using the stochastic programming solver integrated by Lingo. Two models are proposed, Model-I only could solve the problem for a maximum of three periods, due the complexity of using a continuous probability distribution , a second model is proposed with a discretization of the probability distributions (Model-II) which could solve the problem up to four periods. In both models a scenario tree is created. In general, this work compares the efficiency between Model-I and Model-II in resolution time, number of iterations, expected value (EV), wait-and-see value (WS), and expected value of perfect information (EVPI). The obtained results help to determine the advantages about the proposed model (Model-II) with respect to Model I and is useful to understand the scope of both models and in which cases it is advisable to use each one. In addition, both models consider the impact of the service level restriction on the optimal solution and what happen when parameters of the distribution probabilities are varying.

The novelty of this work could be summarized in five points,1) this study provides a mathematical programming model that has been adapted for real needs of a company, which incorporates a service level constraint that it is not found in the literature, usually a confidence percentage is used (which could turn the problem into a chance constraint programming. 2) In the literature only the expected value of the objective function is reported (here and now solution) with the history of the process considered, that is, with the nonanticipativity constraints or the value of the expected objective function. If these constraints are removed, the wait and see solution appears, in our research, both solutions are reported, also the absolute difference of the two solutions is reported, called the expected value of perfect information that could help the company to deal with uncertainty in economic decisions. 3) An extensive sensitivity analysis is carried out, varying the cost parameters, the percentage of the service level and varying the parameters of the probability distribution of uncertainty. Few studies carry out a sensitivity analysis, but in our knowledge, nobody analyzes the impact of service level and varies the parameters of the probability distribution. Through this sensitivity analysis, interesting results were obtained, for example, that the total cost of the APP goes down, when the variability of the production capacity (standard deviation of the probability distribution) is reduced. 4) Due to the complexity of the problem, the software could not solve the problem satisfactorily for a fourth period, finding a solution that is only feasible, then, a second model is developed using discretization of the probability distribution, it has been shown that if the distances of both distributions are minimal, the solution found is closer than the true optimum [2-4] the quality of the proposed model is presented in the results, where both models are compared.

Finally, 5) a methodology to deal with problems using stochastic programming is proposed, although it was applied to the case of this APP, can be implemented in other areas of industrial engineering sciences, such as supply chain networks, problems of vehicle routing, design, and redesign of layouts, among others. Advantages and disadvantages are detailed in the conclusions section. Table 1 shows a comparison between some relevant studies in the area and our study, so that the novelty and contribution of our proposal can be observed.

The contribution of this work is a real problem where uncertainty affects the production system, generally, the models used in the literature consider demand as a random variable with a discrete approximation (one model), in this work, in addition, the human factor is considered as a stochastic parameter that can be modeled and 2 models are compared”. 

Literature review is just providing summary of existing works without any insightful conclusions.

We include a comparative table of the related works in Table 1 and an analysis of this table in the last paragraph “Reviewing the works found that are related to stochastic programming, it was observed that all of them use mixed integer programming, Jamalnia et al. [16], Zhao et al. [18] and Tirkolaee et al. [22] with nonlinear multiobjective, while Kazemi et al. [24] uses multistage stochastic programming which is the same as that used in our proposal. Tirkolaee et al. [22] use demand and costs as stochastic variables, while Kazemi et al. [24] demand and yield; the others, only use a single variable as a stochastic. In their approaches, a single approximation is used to explain the uncertainty, which is discrete, that is, a single stochastic model, and in our case two models, that allow us to compare between the normal distribution and its discretization in order to offer to the company a good solution in a reasonable computational time. The proposals found use the scenerio tree, except Tirkolaee et al. [22] that deal with the problem with weighted goal using GAMS. The level of service is only handled in Zhao et al. [18], but the impact of the service level on the solution is not considered. The sensitivity analysis varying stochastic parameters was not used in the approaches found. After analyzing the characteristics of the studies found, the gap was in considering the level of service, which was a very important restriction for the company in the case study, considering demand and labor as stochastic variables, which were two variables that generate a lot of uncertainty in the company and varying the stochastic parameters. Finally, generate an efficient model, that is to say, a model that obtains a good response in a reasonable computational time, for that reason 2 models were tested for comparison”. 

The structure of the whole paper should be improved.

We structure the paper as Plos One guidelines suggest: (Abstract, Introduction, Materials and methods , Results, Discussion, Conclusions.). In our paper, Material and Methods are the Methodology, the sections are consistent with the steps of figure 2.

Why someone should use your proposed work in practice, and what are the advantages of your work in comparison with others.

The gap founded in the literature that is included in our research is in the last paragraph of the literature review, “The level of service is only handled in Zhao et al. [18], but the impact of the service level on the solution is not considered. The sensitivity analysis varying stochastic parameters was not used in the approaches found. After analyzing the characteristics of the studies found, the gap was in considering the level of service, which was a very important restriction for the company in the case study, considering demand and labor as stochastic variables, which were two variables that generate a lot of uncertainty in the company also varying the stochastic parameters. Finally, generate an efficient model, that is to say, a model that obtains a good response in a reasonable computational time, for that reason 2 models were tested for comparison”. These are advantages of our work, we present a sensitivity analysis of the service level (in percentage), also we include this sensitivity analysis in the stochastic parameters in order to present the enterprise how robust are both models to find a solution in a reasonable computational time. Other industries could use our methodology and also could consider that both models have similar solutions and the discretization of the distributions works more efficient than use the normal distribution.

Summarise the advantages and limitations of the proposed method in practical applications.

The advantages and limitations are in the second paragraph of the conclusions “The advantages offered by the methodology proposed is its flexibility, being able to use it in other problems where there is uncertainty. Some of its limitations, are the need for goodness of fit tests that ensure that the data have a certain probability distribution; if they are not done, the results will be far from optimal Also, the sample size or number of possible realizations of a random variable when it is discretized can make the problem difficult to solve, because the equivalent deterministic grow exponentially as the number of states increases, for this study; solving a fifth state would imply solving a deterministic equivalent of more than two million decision variables and four million constraints because more than 40,000 scenarios are required, which is the maximum number of scenarios that the Lingo software can process without having a computational memory deficit.”

Comparison with existing studies should be added.

We improve all the literature analysis, we add 12 papers of more recent period in the first three paragraphs of the literature review “Considering uncertainty within optimization problems remains a trending topic to be investigated because organizations face to stochastic variables when making decisions. A search was carried out in Scopus and in Web of Science written during 2020 and 2021 that used stochastic programming to solve optimization problems. Huang et. al [5] develop a multistage stochastic optimization model for system operators to efficiently schedule power-generation assets to co-optimize power generation and regulation reserve service under uncertainty. Ghayour et al. [6] present an approach called MLPR with linear programming used as its core in order to solve the influence maximization problem in the linear threshold model, that is one of two classic stochastic propagation models that describe the spread of influence in a network. Robust Multi-product Newsvendor Model with Substitution, where the demand and the substitution rates are stochastic and are subject to cardinality-constrained uncertainty sets that is an NP hard problem is presented in [7].

Also, Basciftci et. al [8] reformulate the robust facility location problem, in which they interpret the moments of stochastic demand as functions of facility-location decisions. In Shone et. al [9], stochastic modeling applications within aviation are presented, with a particular focus on problems involving demand and capacity management and the mitigation of air traffic congestion; using operations research perspective, including analytical queueing theory, stochastic optimal control, robust optimization and stochastic integer programming. Ghasemi et. al [10] present an Evolutionary Learning Based Simulation Optimization (ELBSO) method embedded within Ordinal Optimization. In ELBSO a Machine Learning (ML) based simulation metamodel is created using Genetic Programming (GP) to replace simulation experiments aimed at reducing computation; ELBSO is evaluated on a Stochastic Job Shop Scheduling Problem (SJSSP). Zhang et. al [11], consider a stochastic vehicle routing problem with probability constraints; the probability that customers are served before their (uncertain) deadlines must be higher than a pre-specified target. Wang et. al [12] propose a model to solve a project scheduling problem where resource assignments and activity schedules need to be determined to achieve a set of due-date requirements as well as possible. Torres et. al [13] present multistage stochastic program for the design and management of flexible infrastructure networks with stochastic demands.

 In the methods for solving stochastic programming, Dowson and Kapelevich [14] develop the Julia package for multistage stochastic and dual programming and Gangammanavar et. al [15] work with stochastic decomposition for two-stage stochastic linear programs with random cost coefficients.

Also we include a comparative table of the related works in Table 1 and an analysis of this table in the last paragraph “Reviewing the works found that are related to stochastic programming, it was observed that all of them use mixed integer programming, Jamalnia et al. [16], Zhao et al. [18] and Tirkolaee et al. [22] with nonlinear multiobjective, while Kazemi et al. [24] uses multistage stochastic programming which is the same as that used in our proposal. Tirkolaee et al. [22] use demand and costs as stochastic variables, while Kazemi et al. [24] demand and yield; the others, only use a single variable as a stochastic. In their approaches, a single approximation is used to explain the uncertainty, which is discrete, that is, a single stochastic model, and in our case two models, that allow us to compare between the normal distribution and its discretization in order to offer to the company a good solution in a reasonable computational time. The proposals found use the scenerio tree, except Tirkolaee et al. [22] that deal with the problem with weighted goal using GAMS. The level of service is only handled in Zhao et al. [18], but the impact of the service level on the solution is not considered. The sensitivity analysis varying stochastic parameters was not used in the approaches found. After analyzing the characteristics of the studies found, the gap was in considering the level of service, which was a very important restriction for the company in the case study, considering demand and labor as stochastic variables, which were two variables that generate a lot of uncertainty in the company and varying the stochastic parameters. Finally, generate an efficient model, that is to say, a model that obtains a good response in a reasonable computational time, for that reason 2 models were tested for comparison”. 

Conclusions section has poorly written. Conclusion section should be rewritten by adding advantages, limitations and useful future research directions of the proposed work.

We rewrite the conclusions considering advantages, limitations and future research directions, There are in the second, third fourth paragraphs of the conclusions “The advantages offered by the methodology proposed is its flexibility, being able to use it in other problems where there is uncertainty. Some of its limitations, are the need for goodness of fit tests that ensure that the data have a certain probability distribution; if they are not done, the results will be far from optimal Also, the sample size or number of possible realizations of a random variable when it is discretized can make the problem difficult to solve, because the equivalent deterministic grow exponentially as the number of states increases, for this study; solving a fifth state would imply solving a deterministic equivalent of more than two million decision variables and four million constraints because more than 40,000 scenarios are required, which is the maximum number of scenarios that the Lingo software can process without having a computational memory deficit.

 The study was complemented with sensitivity analysis, in the literature few studies report these analyzes, generally only perform it by varying parameters associated with costs, but in this study is carried out to see the impact of varying the percentage of the policy of level of service, a sensitivity analysis is also carried out, seeing how by varying the parameters of the probability distributions or stochastic parameters (mean and standard deviation) impacts in the solutions and decision variables, so that the company has sufficient information for correct planning in case these parameters could change in the future, or, as an area of opportunity to improve productivity, for example, it could be observed that reducing the variability of the random variable production capacity, that is, reducing its standard deviation, reduces the total cost of the APP.

 Since many of the APPs occupy neither linear functions, a direction for future research is to make a model considering some non-linear functions, such as the inventory cost, also reformulate the problem, removing some restrictions that will allow to have a lower inventory level, allowing to solve the problem in a more efficient way, it is also interested in the use of another algorithm to solve the equivalent determinists, in this work the algorithm B-and-B was used, being able also to use algorithms of cut of plan, consider using multiple kernels in parallel, using multiple heuristics to pre-solve the problem, and using robust algorithms for relaxation of the problem”

References are not consistent.

We made consistent the references using the guidelines of Plos One published in https://journals.plos.org/plosone/s/submission-guidelines#loc-references

6. PLOS authors have the option to publish the peer review history of their article (what does this mean?). If published, this will include your full peer review and any attached files.

Do you want your identity to be public for this peer review? For information about this choice, including consent withdrawal, please see our Privacy Policy.

Reviewer #1: No

Reviewer #2: No

 We use the Preflight Analysis and Conversion Engine (PACE) digital diagnostic tool, https://pacev2.apexcovantage.com/ to ensure that figures meet PLOS requirements.

 Thank you for your time,

Best regards, 

Eva

---

## [Decision Letter · Decision Letter 1]

24 May 2021

Production planning of a furniture manufacturing company with random demand and production capacity using stochastic programming

PONE-D-21-01331R1

Dear Dr. Hernández Gress,

We’re pleased to inform you that your manuscript has been judged scientifically suitable for publication and will be formally accepted for publication once it meets all outstanding technical requirements.

Kind regards,

Dragan Pamucar

Academic Editor

PLOS ONE

Additional Editor Comments (optional):

Reviewers' comments:

Reviewer's Responses to Questions

**Comments to the Author**

1. If the authors have adequately addressed your comments raised in a previous round of review and you feel that this manuscript is now acceptable for publication, you may indicate that here to bypass the “Comments to the Author” section, enter your conflict of interest statement in the “Confidential to Editor” section, and submit your "Accept" recommendation.

Reviewer #1: All comments have been addressed

Reviewer #2: All comments have been addressed

2. Is the manuscript technically sound, and do the data support the conclusions?

Reviewer #1: Yes

Reviewer #2: Yes

3. Has the statistical analysis been performed appropriately and rigorously? 

Reviewer #1: Yes

Reviewer #2: Yes

4. Have the authors made all data underlying the findings in their manuscript fully available?

Reviewer #1: Yes

Reviewer #2: Yes

5. Is the manuscript presented in an intelligible fashion and written in standard English?

Reviewer #1: Yes

Reviewer #2: Yes

6. Review Comments to the Author

Reviewer #1: All the reviewers' comments have been addressed carefully and sufficiently, the revisions are rational from my point of view, I think the current version of the paper can be accepted.

Reviewer #2: The comment of the reviewer has been addressed adequately. So, my recommendation is to accept the paper.

7. PLOS authors have the option to publish the peer review history of their article (what does this mean?). If published, this will include your full peer review and any attached files.

Reviewer #1: No

Reviewer #2: No

---

## [Editor Report · Acceptance letter]

26 May 2021

PONE-D-21-01331R1 

Production planning of a furniture manufacturing company with random demand and production capacity using stochastic programming. 

Dear Dr. Hernández-Gress:

I'm pleased to inform you that your manuscript has been deemed suitable for publication in PLOS ONE. Congratulations! Your manuscript is now with our production department. 

Kind regards, 

on behalf of

Dr. Dragan Pamucar 

Academic Editor

PLOS ONE